# Germline inherited small RNAs facilitate the clearance of untranslated maternal mRNAs in *C. elegans* embryos

Piergiuseppe Quarato [1,2], Meetali Singh [1], Eric Cornes [1], Blaise Li [1,3], Loan Bourdon [1], Florian Mueller[4], Celine Didier [1] & Germano Cecere [1✉]

Inheritance and clearance of maternal mRNAs are two of the most critical events required for animal early embryonic development. However, the mechanisms regulating this process are still largely unknown. Here, we show that together with maternal mRNAs, *C. elegans* embryos inherit a complementary pool of small non-coding RNAs that facilitate the cleavage and removal of hundreds of maternal mRNAs. These antisense small RNAs are loaded into the maternal catalytically-active Argonaute CSR-1 and cleave complementary mRNAs no longer engaged in translation in somatic blastomeres. Induced depletion of CSR-1 specifically during embryonic development leads to embryonic lethality in a slicer-dependent manner and impairs the degradation of CSR-1 embryonic mRNA targets. Given the conservation of Argonaute catalytic activity, we propose that a similar mechanism operates to clear maternal mRNAs during the maternal-to-zygotic transition across species.

[1] Institut Pasteur, Mechanisms of Epigenetic Inheritance, Department of Developmental and Stem Cell Biology, UMR3738, CNRS, 75724, Paris, France. [2] Sorbonne Université, Collège Doctoral, F-75005, Paris, France. [3] Hub de Bioinformatique et Biostatistique - Département Biologie Computationnelle, Institut Pasteur, 75724, Paris, France. [4] Institut Pasteur, Imaging and Modeling Unit, UMR 3691 CNRS, C3BI USR 3756 IP CNRS, 75724, Paris, France. ✉email: germano.cecere@pasteur.fr

The elimination of germline-produced mRNAs and proteins in somatic blastomeres and the concomitant activation of the zygotic genome—the maternal to zygotic transition (MZT)—is critical for embryogenesis[1]. Despite several mechanisms have been discovered to regulate the clearance of maternal mRNAs during the MZT, they only account for the regulation of a small fraction of decayed transcripts, suggesting that factors and pathways responsible for the decay of the majority of maternal mRNAs still remain elusive. In addition, distinct species can adopt different strategies to achieve the clearance of maternal mRNAs. This is why understanding how different species clear maternal mRNAs is fundamental to shed light on additional mechanisms that regulate one of the most important and conserved transitions happening at the beginning of embryogenesis. For instance, small RNAs, such as maternally inherited PIWI-interacting RNAs (piRNAs) in *Drosophila*[2,3] and zygotically transcribed microRNAs (miRNAs) in *Drosophila*, zebrafish, and *Xenopus*[4–6], have been shown to directly regulate the degradation of maternal mRNAs in embryos. However, miRNA functions are globally suppressed in mouse oocytes and zygotes[7,8] and they do not appear to contribute to maternal mRNA clearance in *C. elegans* embryos[9]. This raises the question of whether other type of small RNAs might play a role in this process. In this regard, catalytically active Argonaute and endogenous small interfering RNAs (endo-siRNAs) have been proposed to play an essential role in oocytes and early embryos in place of miRNAs[10,11]. Therefore, in addition to the silencing of repetitive elements, the RNA interference (RNAi) pathway might regulate the removal of maternal mRNAs in animal embryos.

In the nematode *C. elegans*, two waves of maternal mRNA clearance have been documented so far. The first wave of maternal mRNA clearance, regulated by a consensus sequence in the 3′UTR, occurs during the transition from the oocyte to the one-cell embryo[9]. A second wave of clearance occurs in the somatic blastomeres of the developing embryos[12,13]. However, to date, no mechanisms are known to regulate this process in the embryo. Endo-siRNAs produced in the germline are known to be heritable and they can potentially regulate mRNA transcripts in the developing embryos. Two main endogenous small RNA pathways regulate heritable epigenetic processes in the *C. elegans* germline: (1) thousands of piRNAs that target foreign "non-self" mRNAs[14] and (2) endogenous antisense small RNAs that target active "self" mRNAs[15,16]. These antisense small RNAs, called 22G-RNAs, are generated by the RNA-dependent RNA polymerase (RdRP) EGO-1 using target germline mRNAs as a template and are then loaded into the Argonaute CSR-1 (chromosome segregation and RNAi deficient)[16,17]. CSR-1 22G-RNAs are thought to protect germline mRNAs from piRNA silencing and they are essential for fertility and embryonic development[15,18,19]. However, CSR-1 is also responsible for the majority of slicing activity in *C. elegans* extracts and is capable of cleaving complementary mRNAs in vitro[20]. Moreover, CSR-1 slicer activity is capable of fine-tuning some of its mRNA targets in the adult germline[21]. Therefore, whether CSR-1 can protect or degrade its germline mRNA targets is still an open question.

Here, we provide insight into the widespread process of maternal mRNA clearance in early developing embryos. We demonstrate that inherited 22G-RNAs loaded into CSR-1 trigger the cleavage and removal of complementary maternal mRNAs during early embryogenesis. We found that target mRNAs have faster degradation rate due to CSR-1. Moreover, they are depleted of ribosomes, and we provide evidence that the translational status of these maternal mRNAs influences their decay. We employed the auxin-inducible degradation (AID) system[22] to deplete CSR-1 specifically during the MZT in order to study its functions in early embryos. Using this approach, we demonstrate

that the slicer activity of CSR-1 is essential for embryonic viability and is required to post-transcriptionally cleave its embryonic mRNA targets. Given the conservation of Argonaute catalytic activity in metazoans and the conserved function of small RNAs in regulating maternal mRNA clearance in several animal models we propose that a similar mechanism operates to clear maternal mRNAs during the MZT across species.

## Results

### CSR-1 localized to somatic and germline blastomeres and its slicer activity is essential for embryonic development.

To study CSR-1 localization during embryogenesis, we have generated various CRISPR-Cas9-tagged versions of CSR-1 (ref. [23]). CSR-1 is expressed mainly in the germline of adult worms[17] and localizes in the cytoplasm and germ granules (Supplementary Fig. 1a, b). However, in early embryos, CSR-1 localizes to both the germline and somatic blastomeres (Fig. 1a and Supplementary Fig. 1a, b). CSR-1 in the somatic blastomeres is mainly cytoplasmic and persists for several cell divisions (Fig. 1a and Supplementary Fig. 1b). This is in contrast with other germline Argonaute proteins[24,25], such as PIWI, which exclusively segregate with the germline blastomere from the first embryonic cleavage (Fig. 1a and Supplementary Fig. 1b). Animals lacking CSR-1 or expressing a CSR-1 catalytic inactive protein show several germline defects, severely reduced brood sizes, chromosome segregation defects, and embryonic lethality[16,21,26].

To understand the role of CSR-1 during embryogenesis, we depleted CSR-1 protein using the AID system[22]. First, we depleted CSR-1 from the first larval stage (L1), recapitulating the fertility loss and embryonic lethality phenotypes of both *csr-1* (*tm892*) mutant and catalytically inactive CSR-1 mutant worms[21,26] (Fig. 1b, c and Supplementary Fig. 2a, b). Next, we depleted CSR-1 from the late L4 stage (Supplementary Fig. 3a, b). This treatment did not significantly decrease the fertility of adult worms (Fig. 1b), yet all the CSR-1-depleted embryos failed to develop in larvae (Fig. 1c), suggesting CSR-1 plays an essential role during embryogenesis.

To test whether the catalytically active CSR-1 is required for embryonic viability during the MZT, we expressed a single transgenic copy of CSR-1, using MosSCI[27], with a wild-type catalytic domain (DDH) (named CSR-1 WT) or with an alanine substitution of the first aspartate residues within the catalytic DDH motif (ADH) (named CSR-1 catalytic dead), and depleted endogenous CSR-1 from the late L4 larval stage (Fig. 1d). Western blotting analysis showed that the expression of the transgenic CSR-1 WT and the transgenic CSR-1 catalytic dead proteins are similar, even though their levels are lower than the endogenous CSR-1 protein (Supplementary Fig. 3c). Nonetheless, transgenic CSR-1 WT significantly restored embryonic viability (Fig. 1d), whereas the transgenic CSR-1 catalytic dead failed to suppress the embryonic lethality caused by the depletion of endogenous CSR-1 (Fig. 1d). Thus, CSR-1 slicer activity is essential during embryogenesis.

### CSR-1 post-transcriptionally regulates its targets in the embryo.

The localization of maternally inherited CSR-1 in the somatic blastomeres during early embryogenesis suggests that CSR-1 and its interacting 22G-RNAs may regulate specific embryonic targets in somatic cells. To better understand the regulatory function of CSR-1 in the embryo, we identified and compared CSR-1-interacting 22G-RNAs in embryos and adult worms. CSR-1 was immunoprecipitated in embryo populations ranging from 1- to 20-cell stage (Supplementary Fig. 4a) or young adult worms, followed by sequencing of interacting 22G-RNAs. CSR-1-22G-RNA targets in the adult germlines and early

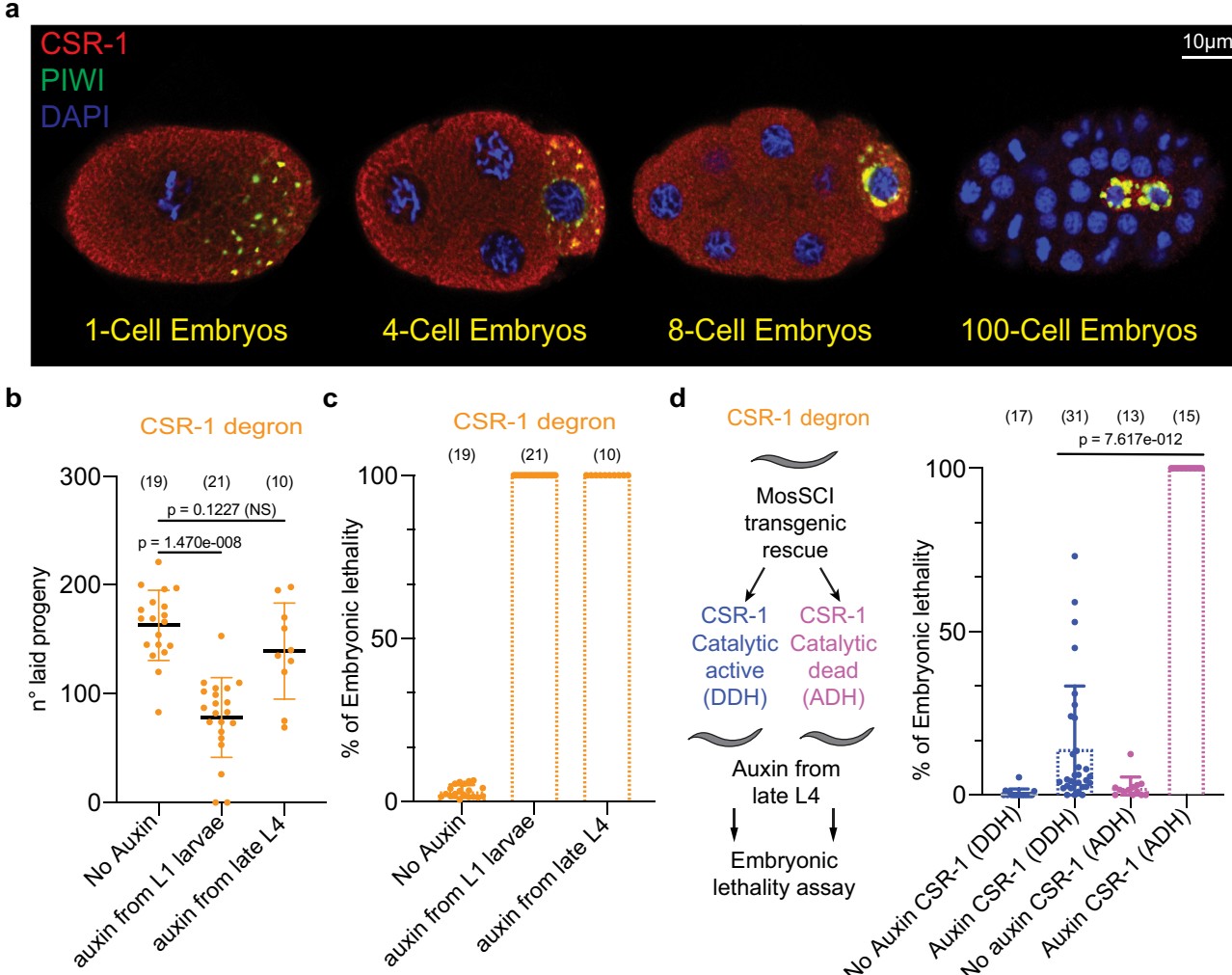

**Fig. 1 CSR-1 localizes to the cytoplasm of somatic blastomeres and its slicer activity is essential for embryonic development. a** Immunostaining of CSR-1 and PIWI in 1-, 4-, 8-, and more than 100-cell embryos of a 3xFLAG::HA::CSR-1 CRISPR-Cas9 strain, using anti-FLAG and anti-PRG-1 antibodies. DAPI signal is shown in blue. Scale bars represent 10 μm. **b** Brood size assays of CSR-1 degron strains with and without Auxin. The data points correspond to the number of living larvae from individual worms. Data are presented as mean ± SD. Two-tailed *P* values were calculated using Mann–Whitney–Wilcoxon tests. The sample size *n* (worms) is indicated in parentheses. **c** Percentages of embryonic lethality from the brood size experiment shown in **b**, measured as the percentage of dead embryos versus the total number among laid embryos. Data are presented as mean ± SD. The sample size *n* (worms) is indicated in parentheses. **d** Percentage of embryonic lethality in CSR-1 degron strains complemented with single-copy transgenic expression of CSR-1 with (ADH) or without (DDH) mutation in the catalytic domain. The percentage of embryonic lethality is calculated by dividing the number of dead embryos by the total number of laid embryos. Dots correspond to the percentages of embryonic lethality from individual worms. Data are presented as mean ± SD. Two-tailed *P* values were calculated using Mann–Whitney–Wilcoxon tests. The sample size *n* (worms) is indicated in parentheses. The graphical representation of the experiment is shown on the left. Source data are available online.

embryos largely overlap (74%) (Fig. 2a). However, the relative abundance of the 22G-RNAs complementary to these targets was different between the adult germlines and early embryos (Fig. 2a, b). Only 29% of CSR-1 targets with the highest levels of complementary 22G-RNAs (>150 reads per million, RPM) in adults and embryos are shared (Fig. 2a, b). Given that CSR-1 cleavage activity on target mRNAs depends on the density of 22G-RNAs[21], our results suggest that CSR-1 may regulate a different subset of mRNAs in the embryos.

To evaluate gene expression changes upon CSR-1 depletion in embryos, we depleted CSR-1 from the late L4 stage and collected early embryo populations fully depleted of CSR-1 (Supplementary Figs. 3a and 4b, c). Analysis of nascent transcription using global run-on sequencing (GRO-seq) revealed that the zygotic transcription in CSR-1 depleted embryos was largely unaffected compared to control untreated embryos (Fig. 2c and Supplementary Fig. 4b), indicating that CSR-1-depleted embryos could initiate zygotic transcription. However, analysis of steady-state mRNA levels using RNA-seq revealed increased levels of CSR-1 embryonic mRNA targets (Fig. 2d and Supplementary Fig. 4c), which correlated with the density of complementary antisense 22G-RNAs loaded onto CSR-1 (Fig. 2d). These results suggest that CSR-1 22G-RNAs in the embryo targets a different set of mRNAs compared to adult germlines and these target mRNAs are directly regulated at the post-transcriptional level in a dose-dependent manner.

**CSR-1 contributes to quicken the degradation of maternal cleared mRNA targets.** In the first phases of embryogenesis, the embryo is transcriptionally silent and relies on maternally inherited mRNA transcripts. Maternal mRNAs degradation is concomitant with the zygotic genome activation, during the

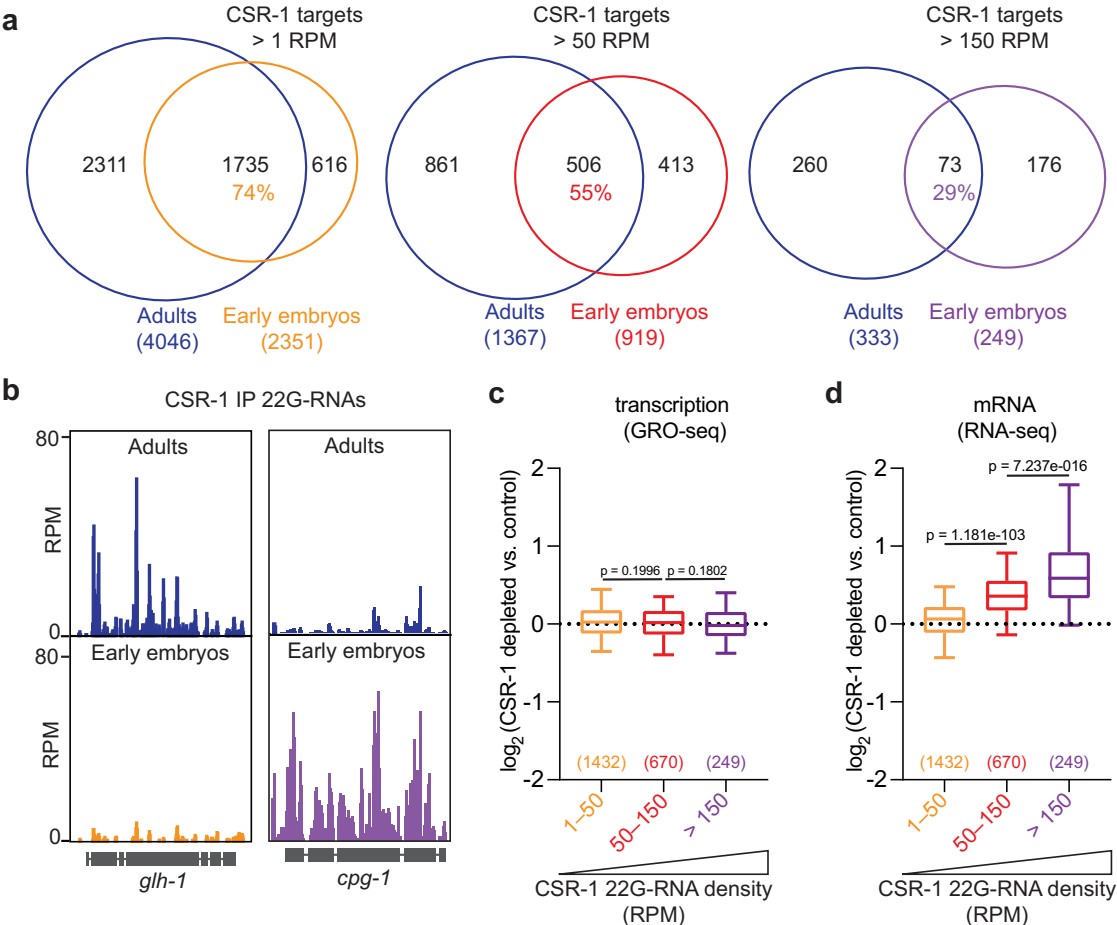

**Fig. 2 CSR-1 post-transcriptionally regulates its targets in the embryo. a** Venn diagram showing the overlap between CSR-1 targets in adults and in early embryos. A gene is considered a CSR-1 target when the ratio between normalized 22G-RNAs from CSR-1 immunoprecipitation and the total RNA in the input is at least 2. We have selected three categories of CSR-1 targets based on the density (reads per million, RPM) of 22G-RNAs from CSR-1 immunoprecipitation: CSR-1 targets >1 RPM, CSR-1 targets >50 RPM, CSR-1 targets >150 RPM. The number of genes and the percentage of overlapping is indicated. **b** Genomic view of two CSR-1 target genes showing normalized 22G-RNAs in CSR-1 IPs from adult (top) or early embryos (bottom). **c, d** Box plots showing the log$_2$ fold change of nascent RNAs (by GRO-seq) or mRNAs (by RNA-seq) in CSR-1-depleted early embryos compared to control for gene sets of increasing 22G RNA density, 1–50 RPM, 50–150 RPM, >150 RPM. The line indicates the median value, the box indicates the first and third quartiles, and the whiskers indicate the 5th and 95th percentiles, excluding outliers. Two-tailed *P* values were calculated using Mann–Whitney–Wilcoxon tests. The sample size *n* (genes) is indicated in parentheses. Source data are available online.

MZT[1]. To gain insight into the type of mRNA targets regulated by CSR-1, we asked whether CSR-1 embryonic targets corresponded to maternal or zygotic mRNA transcripts. We adapted a sorting strategy[28] to obtain embryo populations at the 1-cell stage (99% pure), or enrich for early embryos (from 4- to 20 cells), or late embryos (more than 20 cells) (Supplementary Fig. 4d). Strand-specific RNA-seq was performed on these embryo populations to profile gene expression dynamics. We detected maternal mRNAs from 7033 genes in 1-cell embryos (transcript per million, TPM, >1) (Supplementary Data 1), which we classified into four groups based on their kinetics during embryonic development. "Maternal cleared" mRNAs (1320 genes) corresponded to maternal mRNAs in 1-cell embryos whose levels diminished in early and late stages (Fig. 3a) (Supplementary Data 1). These genes belong to the previously characterized Maternal class II genes, which are maternally inherited in the embryos and degraded in somatic blastomeres[12]. "Maternal stable" mRNAs (1020 genes) corresponded to maternal mRNAs in 1-cell embryos whose levels are stably maintained in early and late embryos (Fig. 3a) (Supplementary Data 1). "Newly Zygotic" mRNAs (704 genes) whose mRNAs accumulated in early and late embryos and were undetectable in 1-cell embryos (TPM < 1)

(Fig. 3a) (Supplementary Data 1), and "Maternal and zygotic" mRNAs (1958 genes) whose mRNAs are inherited in 1-cell embryos (TPM, >1) and their levels increase in early or late embryos (Fig. 3a) (Supplementary Data 1).

Analysis of the embryonic CSR-1 targets revealed that they were inherited in 1-cell embryos and gradually depleted in early and late embryos, similarly to the maternal cleared class of genes (Fig. 3a), suggesting that CSR-1 contributes to maternal mRNA clearance in early embryos. Indeed, the majority of CSR-1 targets were enriched among the category of maternal cleared mRNAs (Supplementary Fig. 5a, b). Next, we analyzed the kinetics of maternal cleared mRNAs that differ in their CSR-1 22G-RNA density. We found that the maternal cleared mRNAs with a high density of CSR-1 22G-RNAs are inherited at higher levels in 1-cell embryos compared to the maternal cleared mRNAs not targeted by CSR-1 (Fig. 3b). Moreover, these mRNA targets show a higher degradation rate (Fig. 3c). These results suggest that in addition to the level of degradation observed for non-target mRNAs, the cleared mRNAs with a high density of CSR-1 22G-RNAs require an additional boost of mRNA degradation to facilitate their clearance in a timely manner. To validate whether CSR-1 is responsible for the faster clearance observed for the

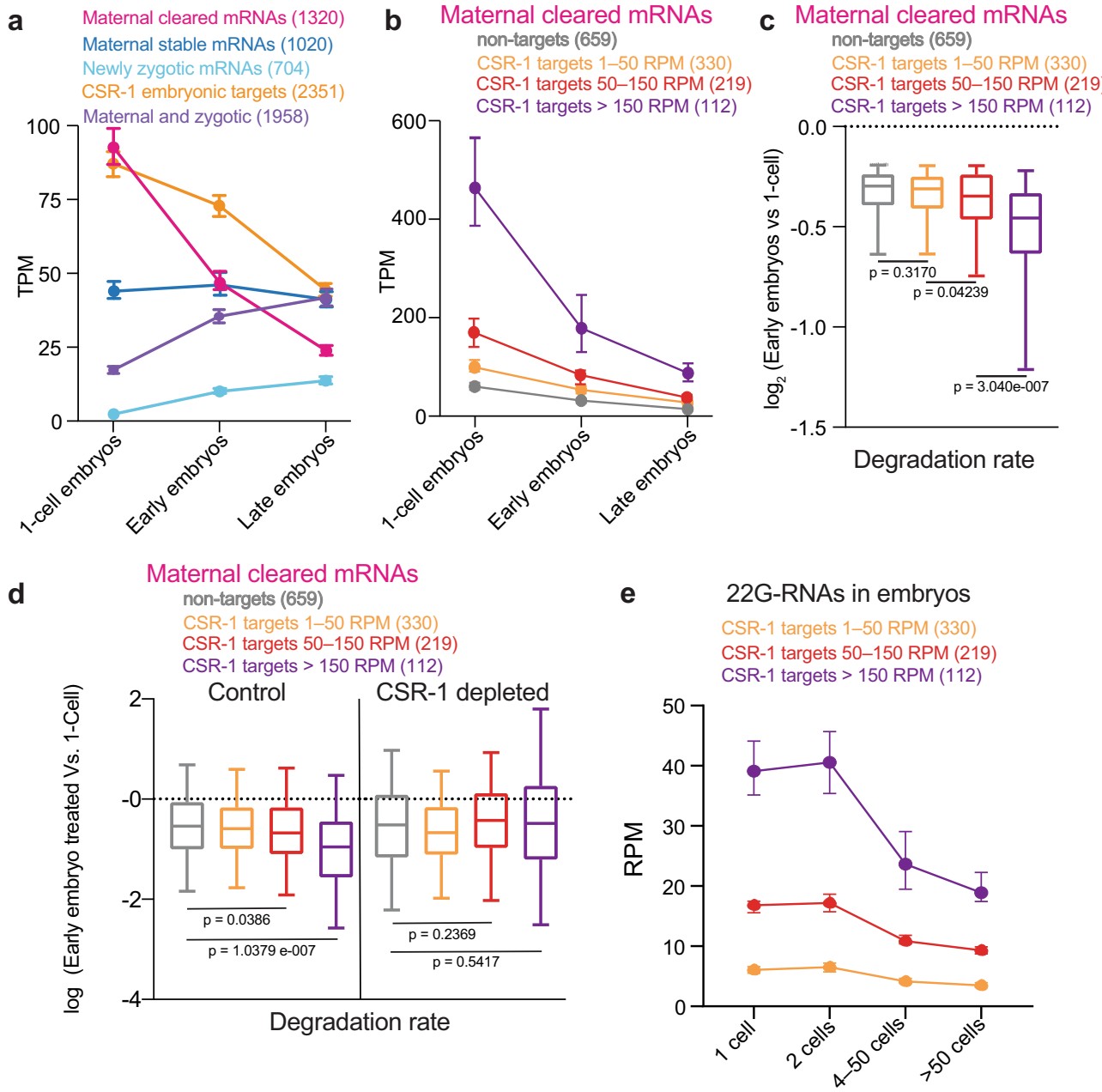

**Fig. 3 CSR-1 contributes to quicken the degradation rate of maternal cleared mRNA targets. a** Detection of maternal cleared mRNAs, maternal stable mRNAs, newly zygotic mRNAs, maternal and zygotic mRNAs, and CSR-1 embryonic target mRNAs by RNA-seq from sorted 1-cell stage, early stage, and late stage embryos. Data are presented as median and 95% confident interval of normalized read abundances in transcript per million (TPM). The sample size $n$ (genes) is indicated in parentheses. **b** Detection of maternal cleared mRNAs for gene sets of increasing 22G RNA density, 1–50 RPM, 50–150 RPM, >150 RPM or non-target genes using RNA-seq normalized reads from sorted 1-cell stage, early stage, and late stage embryos. Data are presented as median and 95% confident interval of normalized read abundances in transcript per million (TPM). The sample size $n$ (genes) is indicated in parentheses. **c** Box plots showing the degradation rate of maternal cleared mRNAs for gene sets of increasing 22G RNA density, 1–50 RPM, 50–150 RPM, >150 RPM or non-target genes. The degradation rate was calculated by the ratio of normalized RNA-seq reads of cleared maternal mRNAs in sorted early embryos versus 1-cell embryos. **d** Box plots showing the degradation rate as in **c**, using normalized RNA-seq reads of cleared maternal mRNAs in control or CSR-1-depleted early embryos versus sorted 1-cell embryos. **e** 22G-RNA levels from sorted embryos enriched in 1-, 2-, 4–50-, and 50-cell stage. Data are presented as median and 95% confident interval of normalized read abundances in reads per million (RPM) from gene sets of increasing 22G RNA density, 1–50 RPM, 50–150 RPM, >150 RPM. In **c, d**, the line indicates the median value, the box indicates the first and third quartiles, and the whiskers indicate the 5th and 95th percentiles, excluding outliers. Two-tailed $P$ values were calculated using Mann–Whitney–Wilcoxon tests. The sample size $n$ (genes) is indicated in parentheses. Source data are available online.

maternal mRNA targets, we measured the degradation rate of maternal cleared mRNAs in CSR-1-depleted embryos. Our analysis shows that the depletion of CSR-1 impaired the higher level of degradation of CSR-1 mRNA targets with a high density of 22G-RNAs compared to non-target mRNAs (Fig. 3d). Finally, we analyzed the dynamics of 22G-RNAs on sorted embryo populations at different stages (Supplementary Fig. 4f). Our analysis shows that the 1-cell embryos are already provided with a

pool of 22G-RNAs antisense to maternal mRNAs and their levels decreased in late embryonic stages (Fig. 3e). Altogether, our results suggest that *C. elegans* embryos inherit a pool of CSR-1 22G-RNAs to target abundant maternal mRNAs that require a higher degradation rate to facilitate their clearance. However, we cannot exclude that the inherited level of CSR-1 22G-RNAs is maintained during early embryogenesis through zygotically produced CSR-1 22G-RNAs.

**CSR-1 cleaves maternal mRNAs in early embryos**. To determine if CSR-1 slicer activity is required for maternal mRNA clearance, we measured gene expression changes in CSR-1-depleted embryos complemented with transgenic expression of CSR-1 catalytic dead or CSR-1 WT (Supplementary Fig. 2c and Supplementary Fig, 2e). Our analysis showed increased accumulation of maternal cleared mRNA targets in slicer-inactive CSR-1 embryos, which correlated with the density of complementary antisense 22G-RNAs loaded onto CSR-1 (Fig. 4a, b). In addition, we cloned and sequenced long (>200 nt) RNAs bearing 5′ monophosphate termini (degradome-seq), which correspond to the end structure generated by Argonaute-mediated cleavage[29], to measure the degradation efficiency (degradome-seq/RNA-seq) of embryonic CSR-1 targets in the absence of CSR-1 catalytic activity. Our results show that the degradation efficiency of CSR-1 targets is reduced in slicer-inactive CSR-1 embryos (Fig. 4c).

To study if CSR-1 is directly regulating these maternal mRNA targets in embryos, we performed RNA-seq on CSR-1 depleted in 1- to 4-cell stage enriched embryos or 4- to 20-cell stage enriched embryos (Supplementary Fig. 6a–c). CSR-1 depletion in 1- to 4-cell embryos did not cause global changes in maternal cleared mRNA target levels (Supplementary Fig. 6d). However, CSR-1 depletion in 4- to 20-cell stage enriched embryos resulted in increased levels of maternal cleared mRNA targets compared to 1- to 4-cell embryos (Supplementary Fig. 6d). Finally, to prove that CSR-1 is degrading mRNAs also in somatic blastomeres we performed RNA single-molecule fluorescence in situ hybridization (smFISH) on a maternal cleared CSR-1 mRNA target in CSR-1-depleted and control embryos. We observed reduced maternal mRNA degradation of the CSR-1 target in somatic blastomeres of 20-cell stage embryos depleted of CSR-1 compared to control (Fig. 4d, e and Supplementary Fig. 6e). Thus, CSR-1 and its interacting 22G-RNAs directly regulate maternal mRNA clearance in somatic blastomeres.

**CSR-1 preferentially targets maternal mRNAs no longer engaged in translation**. To analyze the dynamics of CSR-1-dependent mRNA clearance during embryogenesis, we divided maternal mRNAs (7033) into those degraded early and late during embryogenesis. Early-degraded mRNAs (483 genes) showed at least a twofold reduction in mRNA levels in sorted early embryo populations compared to 1-cell embryos (Fig. 5a and Supplementary Fig. 4d). Late-degraded mRNAs (1572 genes) showed stable levels of mRNAs in sorted early embryo populations compared to 1-cell embryos but were decreased at least twofold in populations of late embryos (Fig. 5a and Supplementary Fig. 4d). Analysis of 22G-RNAs interacting with CSR-1 in the embryo showed that the levels of 22G-RNAs targeting the early-degraded mRNAs were significantly higher compared to late-degraded mRNAs in total RNAs and CSR-1 IPs (Supplementary Fig. 7a, b). Indeed, CSR-1 targets with a high density of 22G-RNAs are highly enriched for early-degraded mRNAs in contrast to late-degraded mRNAs (Supplementary Fig. 7c). In addition, the level of 22G-RNAs antisense to late-degraded mRNAs did not increase during

late embryogenesis (Supplementary Fig. 7d). These results together with the observation that CSR-1 is absent in somatic cells during late embryonic development (Supplementary Fig. 1b) suggest that CSR-1 might preferentially regulate early-degraded mRNAs.

Next, we generated metaprofiles for the levels of CSR-1-interacting 22G-RNAs levels along the gene body of early- and late-degraded mRNA targets (Fig. 5b, c). The early-degraded mRNA genes showed an enrichment of CSR-1-loaded 22G-RNAs at the 3′ end of the genes as well as along the whole coding sequence (Fig. 5b, d). The late-degraded mRNA genes instead showed enrichment of CSR-1-loaded 22G-RNAs mainly at the 3′ end, corresponding to the 3′UTR (Fig. 5c, d). Previous reports have shown that the rate of mRNA clearance during MZT is influenced by translational efficiency[30,31]. Thus, the different translational status of early- and late-degraded mRNAs may influence the accessibility of CSR-1-loaded 22G-RNAs antisense to the coding sequences of cleared mRNAs. To test this hypothesis, we measured ribosomal occupancy by Ribo-seq in a population of early embryos (Supplementary Fig. 7e). Early-degraded mRNAs showed significantly lower ribosomal occupancy and translational efficiency compared to late-degraded mRNAs (Fig. 5e, f). These results suggest that ribosomal loading on maternal mRNAs can influence the accessibility of CSR-1 slicer activity.

**Ribosome occupancy affects CSR-1-mediated maternal mRNA clearance**. Previous studies have shown that mRNA codon usage can influence translational efficiency and maternal mRNA decay during the MZT[30,31]. However, the early- and late-degraded mRNAs did not show any differences in their codon usage (Supplementary Fig. 7f, g). Because the translation of germline mRNAs is primarily regulated by their 3′UTR in *C. elegans*[32], we investigated whether different 3′UTRs from early- and late-degraded mRNAs can impact translation and mRNA degradation in early embryos. We generated two single-copy transgenic lines expressing a germline *mCherry::h2b* mRNA reporter with either a 3′UTR from an early- (*egg-6* 3′UTR) or a late- (*tbb-2* 3′UTR) degraded mRNA target (Fig. 6a). The distribution of endogenous CSR-1-bound 22G-RNAs from these two genes resembled the features of early- and late-degraded mRNAs, having a different level of 22G-RNAs mapping on the coding sequence and a similar level on the 3′UTR (Supplementary Fig. 8a). The expression of the two *mCherry::h2b* mRNA reporters was similar in adult worms (Fig. 6b). However, the level of *mCherry::h2b* mRNA fused with the early-degraded *egg-6* 3′UTR was lower in embryo, indicating increased degradation in early embryos (Fig. 6b). To verify that the different 3′UTRs affect the translational efficiency of *mCherry::h2b* mRNA in embryos we quantified the ribosome occupancy by Ribo-seq and mRNA level by RNA-seq (Supplementary Fig. 8b–d), and observed decreased translational efficiency of *mCherry* transgenic reporter with *egg-6* 3′UTR (Fig. 6c). To test whether CSR-1 22G-RNAs contribute to the increased decay of the *mCherry::h2b* reporter fused to the *egg-6* 3′UTR, we sequenced small RNAs loaded onto CSR-1 in early embryo populations from the two transgenic lines (Supplementary Fig. 9a). We observed increased levels of total 22G-RNAs (Supplementary Fig. 9b) and CSR-1-bound 22G-RNAs on the coding sequences of the *mCherry* transgenic reporter with *egg-6* 3′UTR compared to the *tbb-2* 3′UTR reporter (Fig. 6d, e and Supplementary Fig. 9c). Given that ribosome translation antagonize CSR-1 22G-RNA biogenesis and targeting in adult worms[33], our results suggest that the translation level of maternal mRNAs can influence the rate of CSR-1-mediated mRNA cleavage in early embryos.

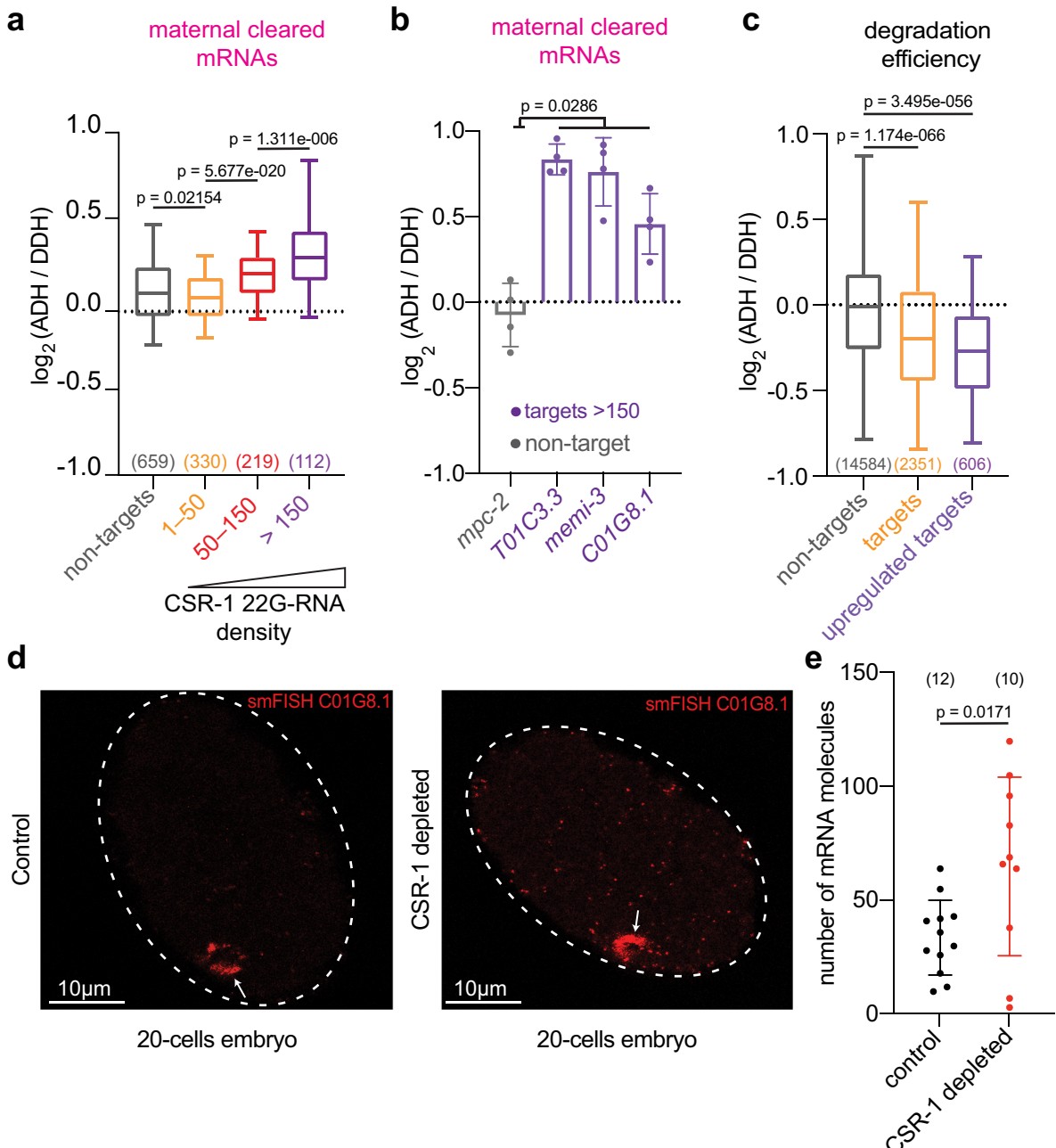

**Fig. 4 CSR-1 cleaves maternal mRNAs in early embryos. a** Box plots showing the $\log_2$ fold change of cleared maternal mRNAs in CSR-1-depleted early embryos rescued with transgenic expression of CSR-1 ADH (catalytic dead) or CSR-1 DDH (catalytic active) for gene sets of increasing 22G RNA density, 1–50 RPM, 50–150 RPM, >150 RPM or non-target genes. The line indicates the median value, the box indicates the first and third quartiles, and the whiskers indicate the 5th and 95th percentiles, excluding outliers. The sample size $n$ (genes) is indicated in parentheses. **b** RT-qPCR showing $\log_2$ fold change of maternal cleared mRNAs in CSR-1-depleted embryos with transgenic expression of CSR-1 with (ADH) or without (DDH) mutation in the catalytic domain. Data are presented as mean ± SD from four biologically independent replicates. **c** Box plots showing the $\log_2$ fold change of the degradation efficiency in CSR-1-depleted early embryos rescued with transgenic expression of CSR-1 ADH versus CSR-1 DDH. The degradation efficiency was calculated by the ratio of normalized degradome-seq reads versus RNA-seq reads. The line indicates the median value, the box indicates the first and third quartiles, and the whiskers indicate the 5th and 95th percentiles, excluding outliers. The sample size $n$ (genes) is indicated in parentheses. Genes with 0 counts in RNA-seq (ADH/DDH) have been excluded from calculation. "Upregulated targets" = CSR-1 targets with log2FC (ADH/DDH) > 0 and adjusted $P$ value <0.05 by RNA-seq. **d** Representative images of *C01G8.1* mRNA smFISH in control (left) and CSR-1 depleted (right) 20-cell stage embryos. Arrows indicates the accumulation of mRNAs in germline blastomere. **e** smFISH quantification of *C01G8.1* mRNA (embryonic CSR-1 target) in somatic blastomeres of control (black) and CSR-1 depleted (red) 20-cell stage embryos. Data are presented as mean ± SD and the dots are the number of mRNA molecules per embryo section. The sample size $n$ (embryos) is indicated in parentheses. Scale bars represent 10 μm. In **a**, **b**, **c**, **e**, two-tailed $P$ values were calculated using Mann–Whitney–Wilcoxon tests. Source data are available online.

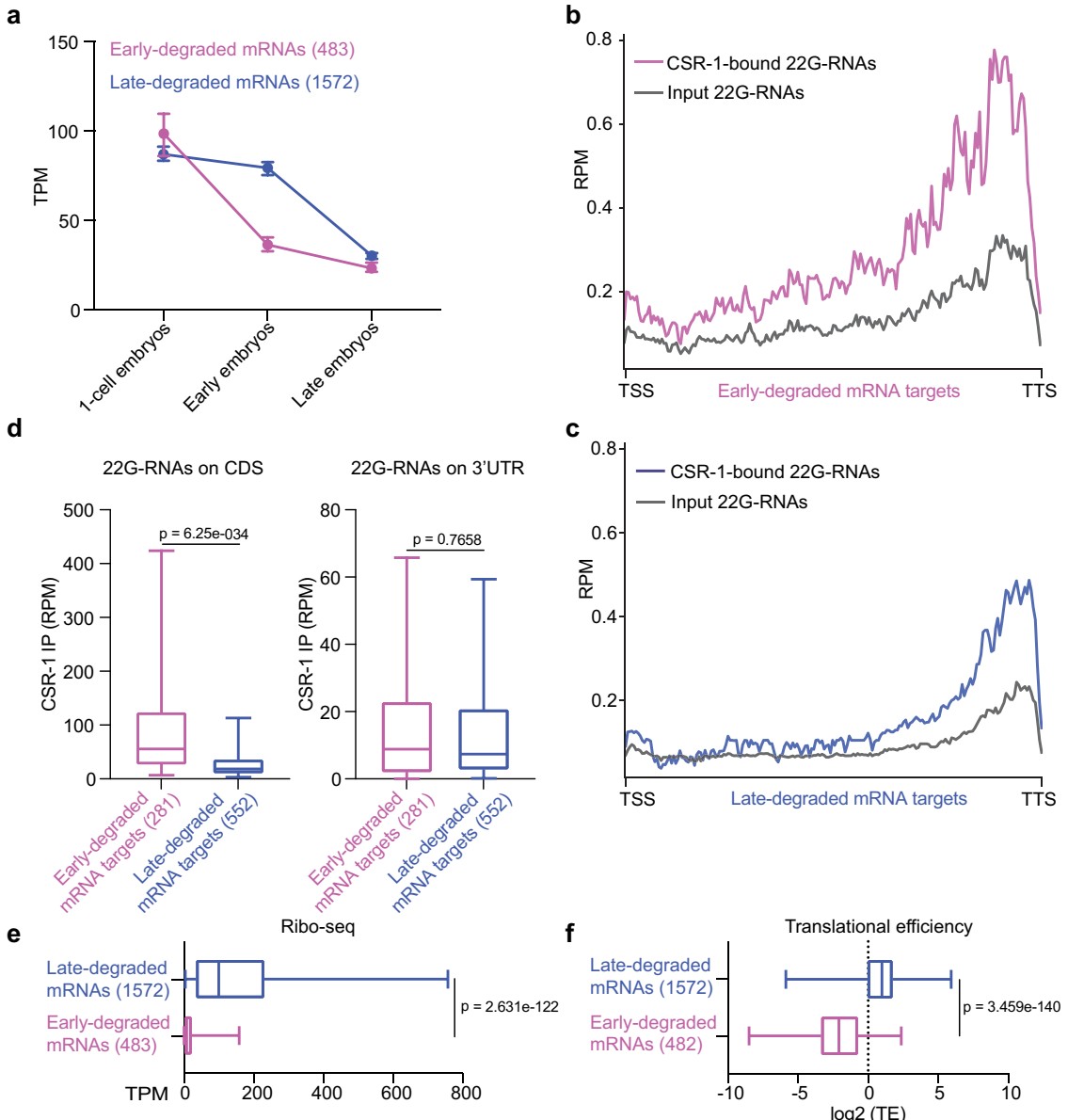

**Fig. 5 CSR-1 preferentially targets maternal mRNAs no longer engaged in translation. a** Detection of early- and late-degraded mRNAs by RNA-seq from sorted 1-cell, early and late embryo stages. Data are presented as median and 95% confident interval of normalized read abundances in transcript per million (TPM). The sample size *n* (genes) is indicated in parentheses. **b, c** Metaprofile analysis showing normalized 22G-RNA reads (RPM) across early-(magenta) and late-degraded (blue) mRNA targets in CSR-1 immunoprecipitation (IP) or total RNA input (gray). The transcriptional start site (TSS) and transcriptional termination site (TTS) is indicated. The average from three biologically independent replicates is shown. **d** Box plots showing the abundance of CSR-1-bound 22G-RNAs antisense to coding sequence (CDS) or 3'UTR of early- and late-degraded mRNAs. **e** Box plots showing the ribosomal occupancy (by Ribo-seq) of early- and late-degraded mRNAs in early embryos. **f** Median levels and 95% CI of translational efficiency (TE) of early- and late-degraded mRNAs in early embryos. In **d–f**, the line indicates the median value, the box indicates the first and third quartiles, and the whiskers indicate the 5th and 95th percentiles, excluding outliers. The sample size *n* (genes) is indicated in parentheses. Two-tailed *P* value were calculated using the Mann–Whitney–Wilcoxon test. Source data are available online.

In summary, our study identifies a function of the conserved Argonaute slicer activity in the degradation of maternally inherited mRNAs during the MZT.

## Discussion

In this study, we have shown that the maternally inherited CSR-1 protein and its interacting 22G-RNAs trigger the cleavage of hundreds of complementary maternal mRNA targets in early embryos. Despite most of the inherited Argonautes localize to the germline blastomeres, the inherited CSR-1 also localizes to the cytoplasm of somatic blastomeres for several cell divisions during early embryogenesis. We have shown that in *C. elegans* embryos CSR-1 is loaded with 22G-RNAs antisense to mRNAs that undergo rapid degradation during early embryogenesis, and CSR-1 slicer activity facilitates the clearance of these maternal mRNAs. We have observed that the maternal mRNAs targeted by CSR-1 are inherited at higher levels in 1-cell embryos compared to the other maternal cleared mRNAs. Nonetheless, CSR-1 targets show a faster degradation rate, which depends on maternal inherited CSR-1. We have also observed that CSR-1 is preferentially loaded with 22G-RNAs antisense to the coding sequences of untranslated

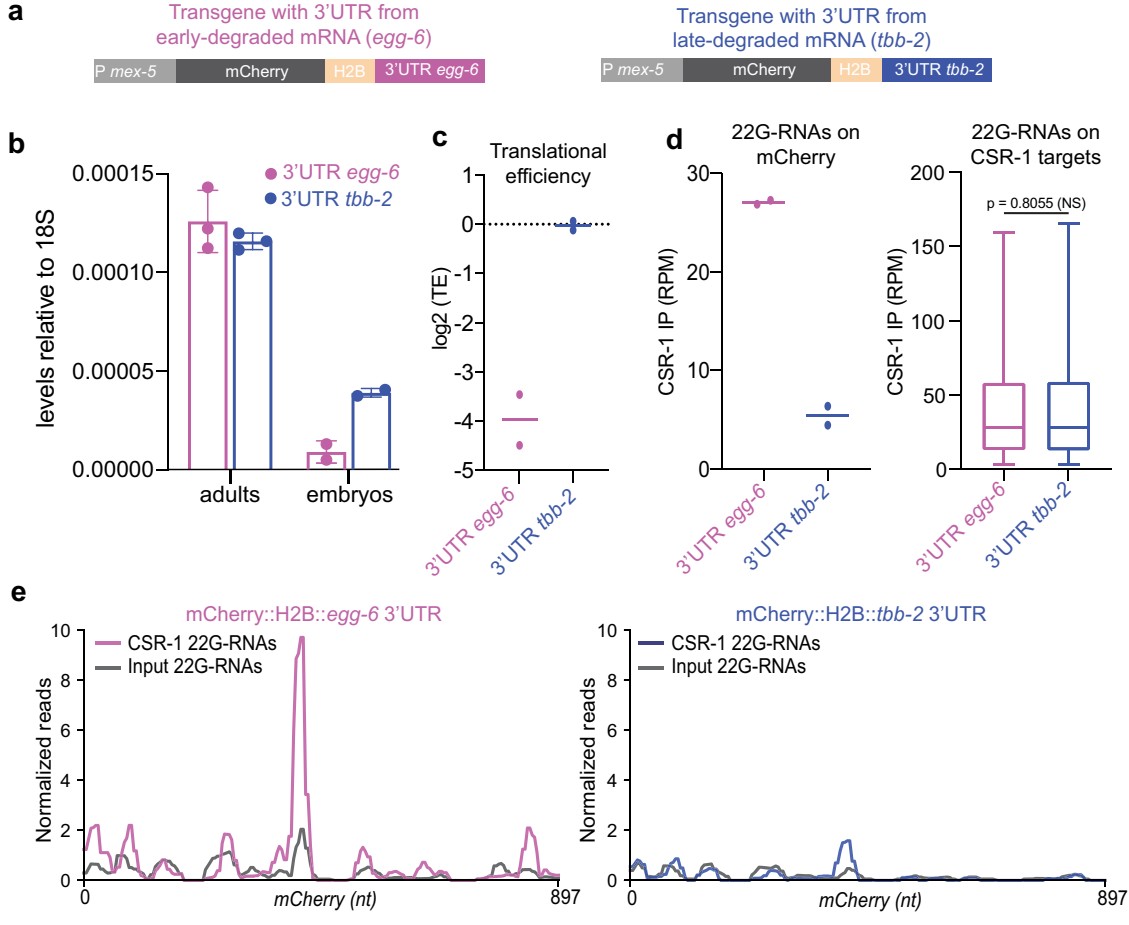

**Fig. 6 Ribosome occupancy affects CSR-1-mediated maternal mRNA clearance. a** Schematic of the germline-expressed single-copy *mCherry::h2b* transgenic mRNA fused to a 3′UTR derived from an early- (*egg-6* 3′UTR) or a late-degraded (*tbb-2* 3′UTR) mRNAs. **b** RT-qPCR assay to detect the levels of mCherry mRNA from the two transgenic reporters in adults and early embryos. The bars indicate the mean, the error bars the standard deviation, and the black dots individual data from three or two biologically independent replicates in adults and embryos respectively. The levels of mCherry mRNA are normalized on the levels of ribosomal 18S RNA. **c** Translational efficiency (TE) of *mCherry::h2b* transgenic mRNAs. The lines indicate the mean and the dots individual data from two biologically independent experiments. **d** Abundance of CSR-1-bound 22G-RNAs antisense to mCherry (left) or the coding sequence of embryonic CSR-1 targets (right). The lines indicate the mean and the dots individual data from two biologically independent experiments (left). For the box plots (right), the line indicates the median value, the box indicates the first and third quartiles, and the whiskers indicate the 5th and 95th percentiles, excluding outliers. Two-tailed *P* value were calculated using the Mann–Whitney–Wilcoxon test. Sample size *n* = 2351 (CSR-1 mRNA targets). **e** Metaprofile analysis showing normalized 22G-RNA reads (RPM) across *mCherry* nucleotide sequence from *mCherry::h2b* transgenic mRNA fused to *egg-6* 3′UTR (left) or *tbb-2* 3′UTR (right) in CSR-1 immunoprecipitation (IP) or total RNA input. The average from two biologically independent replicates is shown. Source data are available online.

mRNA targets in embryos. Therefore, we propose that CSR-1 slicer activity is required to quicken the degradation of abundant maternal mRNAs that are no longer translated during early embryogenesis.

**Translation tunes 22G-RNAs levels regulating the timing of CSR-1 slicing.** Based on our results, we speculate that active translation prevents the accessibility of CSR-1 22G-RNAs on complementary mRNA targets and hence the slicing through the coding sequences of translated maternal mRNAs. Therefore, the regulation of translation of maternal mRNAs ensures that CSR-1 22G-RNA complexes can remove these mRNAs only when they are not translated. We have also shown that by modulating the translational activity of a maternal mRNA sequence using different 3′UTRs, we could detect different amounts of antisense 22G-RNAs. Even if we cannot exclude that different features in the 3′UTRs used can influence the production of 22G-RNAs, we speculate that the presence of ribosomes on the coding sequence of maternal mRNAs can antagonize the synthesis of 22G-RNAs by

the RdRP. Previous studies suggested that the synthesis of 22G-RNAs by the RdRP EGO-1 starts from the 3′ end of protein-coding transcripts[34]. We have also recently observed that CSR-1 slicing is required for the production of 22G-RNAs targeting the gene body and this activity is antagonized by translation[33]. However, CSR-1 slicing is not required for the production of 22G-RNAs targeting the 3′UTR, which depends on the RdRP EGO-1 (ref. [33]). Here we found that translated late-degraded maternal mRNAs are devoid of 22G-RNAs targeting the gene body and they only have 22G-RNAs targeting their 3′UTR. On the contrary, untranslated early-degraded maternal mRNAs are targeted by 22G-RNAs along the whole transcript, including the gene body. Based on our findings, we propose that the presence of ribosomes on the coding sequence of late-degraded RNAs can antagonize CSR-1 slicer activity and 22G-RNA synthesis on the gene body to protect late-degraded maternal mRNAs from degradation. On the contrary early-degraded maternal mRNAs are readily able to be degraded by inherited CSR-1-bound 22G-RNAs. Therefore, ribosome translation would act as a molecular clock that

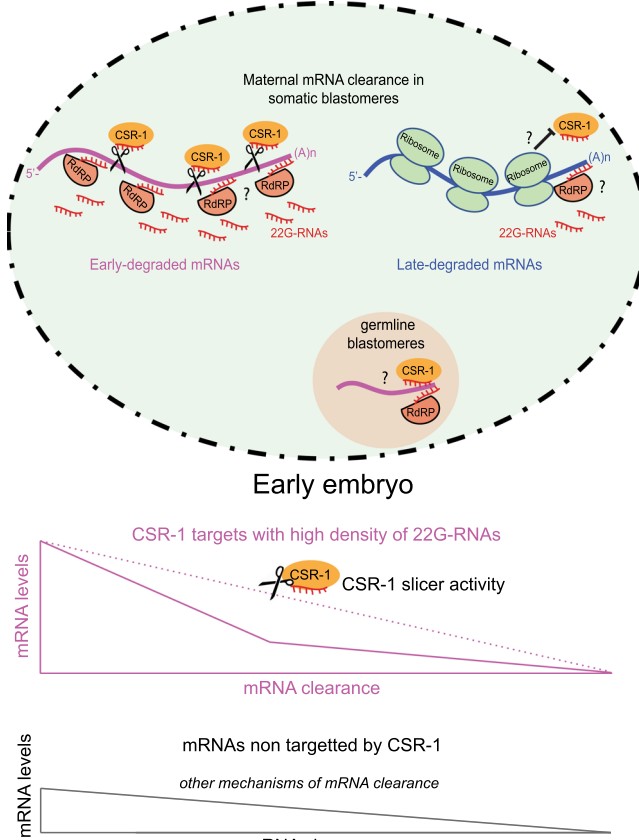

**Fig. 7 Model illustrating CSR-1-dependent clearance of maternal mRNA in *C. elegans* embryos.** During early embryogenesis, maternally inherited CSR-1 protein targets untranslated early-degraded maternal mRNAs for slicer-dependent degradation in somatic blastomeres. The presence of ribosomes on translated late-degraded maternal mRNAs might prevent CSR-1 accessibility and accumulation of new 22G-RNA antisense to the gene body. RNA in the germline blastomere might be protected from CSR-1 slicer activity. A fraction of maternal cleared RNAs not targeted by CSR-1 might be degraded by yet uncharacterized mechanisms. The CSR-1 pathway is an additional inherited mechanism that contributes to quicken the degradation of abundantly inherited mRNAs cleared during early embryogenesis.

determines the timing of CSR-1-dependent degradation of maternal mRNAs during early embryogenesis (Fig. 7). However, how late-degraded maternal mRNAs are degraded in late phases of embryogenesis remains unknown. One possibility would be that once late-degraded maternal mRNAs are not engaged in translation, during late embryogenesis, they would become accessible for CSR-1 slicer activity and RdRP EGO-1 to produce zygotic 22G-RNAs. However, it is important to note that CSR-1 protein is also degraded at around 100-cell stage, where it is only present in the primordial germ cells Z2/Z3 (Fig. 1a and Supplementary Fig. 1b). Moreover, 22G-RNAs antisense to late-degraded mRNAs do not increase during embryogenesis (Supplementary Fig. 7d). For these reasons, CSR-1-mediated degradation of maternal transcripts can also be temporarily regulated and restricted to the early phase of embryogenesis. Thus, we cannot exclude that other unknown mechanisms might participate in the degradation of late-degraded maternal mRNAs in somatic blastomeres.

**CSR-1 slicing in adults and early embryos.** CSR-1 has been extensively studied in the adult germlines, where it has been proposed to protect active genes from piRNA silencing[18,19],

promote germline transcription[35,36], regulate the biogenesis of histone mRNAs[37], and fine-tune germline mRNAs loaded into oocytes[21]. The catalytic activity of CSR-1 and its gene regulatory function on target mRNAs has been proven to be essential for fertility and chromosome segregations[21]. Similarly, our work provided evidence that CSR-1 slicer activity is essential for embryonic viability independently from its germline functions. Therefore, we propose that the accumulation of maternal mRNAs in embryos lacking CSR-1 slicer activity might be deleterious for the developing embryo.

Based on our findings we conclude that CSR-1 catalytic activity regulates its targets both in adult germlines and in early embryos. However, the effect on targeted mRNA is different in the two developmental stages. Indeed, while CSR-1 activity only results in a fine tuning of its germline targets, CSR-1 embryonic targets are cleared during early embryogenesis. Importantly, embryos are transcriptionally silent for the first phases of embryogenesis, while CSR-1 targets are also actively transcribed in the *C. elegans* germline. Therefore, we propose that the CSR-1 slicer activity combined with the differential transcriptional status of CSR-1 targets determines the different gene regulatory outcomes in adults and embryos. Moreover, we have recently reported that CSR-1 slices mRNA targets in the cytosol and not in the germ granules of adult worms[33]. Therefore, the enrichment of CSR-1 in adult germ granules might further contribute to preventing CSR-1 catalytic activity, which is fully released in the cytosol of somatic blastomeres in embryos.

Maternally inherited mRNAs, including CSR-1 embryonic targets, also accumulates in germline blastomeres, where they colocalize with embryonic germ granules[12,25]. CSR-1 protein is also expressed in germline blastomeres, but we were unable to evaluate whether CSR-1 is also cleaving mRNAs in these cells. However, CSR-1 catalytic activity is only acting on cytosolic mRNAs[33], suggesting that germ-granule localized mRNAs might not be regulated by CSR-1 catalytic activity in embryos. This is also in agreement with the notion that maternal mRNAs localized in germ granules are protected from degradation[12,25].

**Are 3′UTRs directly involved in the regulation of maternal mRNA decay?** It is known that translation affects maternal mRNA clearance, and recent works have shown how the codon usage of maternal mRNAs regulates translation and degradation rates in zebrafish embryos[30,31]. Here we show that in *C. elegans* embryos early- and late-degraded maternal mRNAs do not differ in their codon usage. Instead, we have observed that the 3′UTR is sufficient to regulate the translational efficiency of the coding sequence of a maternal mRNA in embryos, and in turn its degradation rate, including CSR-1 slicer activity. Even if we implicate CSR-1 slicer activity in the degradation of early-degraded mRNAs, we cannot exclude that sequences or other features in the 3′UTR can directly regulate CSR-1 22G-RNA biogenesis and the time of decay observed in early- versus late-degraded maternal mRNAs during MZT. A consensus sequence in the 3′UTR of some maternal mRNAs has been previously shown to play a role in the degradation of mRNAs from oocytes to 1-cell embryos in *C. elegans*[9]. Therefore, it would be interesting in future studies to identify other consensus sequences located in the 3′UTR of maternal mRNAs and also RNA-binding proteins that might contribute together with CSR-1 to regulate mRNA clearance during embryogenesis.

**Endo-siRNAs may have a conserved role during MZT in metazoans.** The requirement of Argonaute slicer activity to degrade endogenous target mRNAs has only been described in the silencing of repetitive elements in animals. Here, we propose

that the RNAi activity of CSR-1 on maternal mRNAs might be a conserved mechanism required for the MZT and mRNA clearance across species. Indeed, catalytically active Argonaute proteins are conserved in animals, including humans. For example, of the eight human Argonaute proteins, AGO2 and AGO3 have been shown to possess a catalytic activity able to cleave complementary mRNA targets[38–40].

Endogenous small RNAs targeting protein-coding genes, similar to the one loaded onto CSR-1, have been detected in mouse oocytes and embryonic stem cells[41–43]. This suggests that mammalian RNAi, in addition to the role in suppressing the expression of repetitive elements, might also regulate endogenous genes. Given that the inheritance of maternal catalytically active Ago2 is essential for oocyte[10] and early embryonic development[11] in mouse, the current hypothesis is that Ago2 can promote the degradation of maternal mRNAs in embryos[44]. Strikingly, catalytically inactive Ago2 in mouse oocytes causes chromosome segregation and spindle assemble defects[10] similar to CSR-1 catalytic mutant[21], suggesting they might regulate common functional mRNA targets during oogenesis. Ago2 also loads miRNAs in mouse. Even if miRNAs contribute to regulate maternal mRNA clearance in some animal models, including *Drosophila*, zebrafish, and *Xenopus*, their functions are suppressed in mouse oocytes[7,8]. Therefore, Ago2 and endogenous small RNAs might play a role in maternal mRNA clearance during the MZT in mouse.

The role of miRNAs in maternal mRNA clearance in *C. elegans* is also unclear[9], despite ALG-1 and ALG-2, the two catalytically competent Argonaute protein loading miRNAs, are essential for embryonic development[45–47]. The *C. elegans* PIWI protein PRG-1 has also a slicer catalytic activity[48]. However, PRG-1 segregates with the germline lineage from the first embryonic cleavage and is excluded from somatic blastomeres. Therefore, it is unlikely that *C. elegans* piRNAs could participate in maternal mRNA clearance as occurs in *Drosophila* and *Aedes*[2,49].

Based on our findings, we propose that endogenous small RNAs and Argonaute-mediated cleavage of mRNA may be a conserved mechanism in metazoans to degrade maternal mRNAs in oocyte and embryos.

## Methods

**_C. elegans_ strains and maintenance**. Strains were grown at 20 °C using standard methods[50]. The wild-type reference strain used was Bristol N2. A complete list of strains used in this study is provided in Supplementary Data 2.

### Generation of transgenic animals

*Generation of CRISPR-Cas9 lines*. CRISPR-Cas9 alleles were generated similar to as described in ref. [23]. We used a *mCherry::3×Flag::ha::csr-1* (ref. [23]) as an entry strain to introduce an auxin-inducible Degron tag and obtain a *Degron::mCherry::3xFlag::ha::csr-1* strain suitable for the AID of CSR-1, after crossing with CA1352, a strain carrying a single copy of the germline-expressed TIR-1 protein[22]. CRISPR-Cas9 guide RNA sequences are listed in Supplementary Data 3.

*Generation of MosSCI lines*. Strains carrying a single copy insertion of *mex-5P::gfp::csr-1::tbb-2 3′UTR* or *mex-5P::gfp::csr-1(ADH)::tbb-2 3′UTR* were generated by MosSCI[51] and the Gateway Compatible Plasmid Toolkit[51]. The sequences of *csr-1* and *csr-1 (ADH)* were amplified from genomic DNA by PCR from strains carrying either a wild-type copy of *csr-1* or *csr-1(ADH)* and cloned in pENTR-D-TOPO (Invitrogen). Plasmids used for Multisite Gateway reaction are listed as follows: pJA245, pCM1.36, and pCFJ212. Strain EG6703 was used as an entry strain for single insertion in chromosome IV.

*3′UTR replacement experiment*. A sequence of 800 bp downstream of the stop codon of egg-6 was amplified from genomic DNA by PCR and used as 3′UTR. We used the single-copy *transgene mCherry::his-11::tbb-2 3′UTR* as entry strain to insert *egg-6 3′UTR* by CRISPR-Cas9 as described above. *egg-6 3′UTR* was inserted right after the STOP codon of *his-11*. The specific expression of the inserted UTR was validated by reverse transcriptase quantitative PCR (RT-qPCR).

**Immunostaining**. Gonads were dissected on PBS, 0.1% Tween-20 (PBST) containing 0.3 mM levamisole on 0.01% poly-lysine slides. Samples were immediately freeze cracked on dry ice for 10 min and fixed at −20 °C for 15 min in methanol, and 10 min in acetone. Blocking was performed for 30 min at room temperature. Primary antibody was incubated overnight at 4 °C in PBS, 0.1% Tween-20, 5% BSA. The secondary antibody was incubated for 1 h at room temperature. Washes were performed with PBS, 0.1% Tween-20 (PBST). DNA was stained with DAPI. For embryo immunostaining, a drop of embryos from bleached adults was used instead of gonads. Images of adult germlines and embryos were acquired on a Zeiss LSM 700 confocal microscopy. The primary antibodies used were anti-FLAG (Sigma, F1804) and anti-PRG-1 (gift from the Mello Lab) antibodies at a dilution of 1:500 and 1:800, respectively, and the secondary antibodies used were goat anti-mouse Alexa Fluor 488 (Invitrogen, A11001) and goat anti-rabbit Alexa Fluor 568 (Invitrogen A11011) antibodies at a dilution of 1:500.

**Auxin-inducible depletion of CSR-1**. Auxin-inducible depletion has been performed as described in ref. [22]. In all experiments, 250 mM auxin stock solution was prepared in ethanol and stored at 4 °C. Auxin plates or ethanol plates were prepared by the addition of auxin or only ethanol to NGM plates (final concentration: 500 µM auxin, 0.5% ethanol for auxin plates and 0.5% ethanol for ethanol plates). Plates were seeded with OP50 *Escherichia coli*, stored at 4 °C, and warmed at room temperature before the experiment. Worms were placed on auxin or ethanol plates from L1 or from late L4 stage as explained for each experiment.

### Brood size assays

*WT and CSR-1 mutants*. Single L1 larvae were manually picked and placed onto NGM plates seeded with OP50 *E. coli* and grown at 20 °C until adulthood and then transferred on a new plate every 24 h for a total of two transfers. The brood size of each worm was calculated by counting the number of embryos and larvae laid on the three plates. For the auxin-depleted experiments NGM plates were supplemented with ethanol (control) or auxin.

*Brood size of CSR-1 auxin-depleted worms during late L4*. For CSR-1 depletion during late L4: brood size assay has been performed as described above with the exception that worms were grown from L1 on regular NGM plates and transferred onto auxin or ethanol plates 44 h after hatching and transferred on new corresponding Auxin and Ethanol plates every 24 h for a total of 2 transfers.

*Embryonic lethality assay of CSR-1 rescue experiments*. Single L1 larvae of strains carrying Degron::CSR-1 complemented with the single-copy insertion Mex-5P:: GFP::CSR-1::tbb-2 3′UTR or Mex-5P::GFP::CSR-1(ADH)::tbb-2 3′UTR were manually picked and placed onto NGM plates seeded with OP50 *E. coli* and grown at 20 °C until adulthood. Since transgene was prone to silencing, adult worms were allowed to lay at least 65 embryos and then adults were removed from the plates and imaged to check for GFP::CSR-1 expression. Only plates from GFP-expressing adults were used for the assay. The percentage of embryonic lethality is calculated by dividing the number of dead embryos for the total number of laid embryos.

**Live imaging**. Image in Supplementary Fig. 1a (4-cell embryo) was acquired using a Plan-Apochromat ×100/1.46 Oil DIC M27 objective on a Zeiss Axio Imager.M2 equipped with a PIXIS 1024 CCD camera (Princeton Instruments). Images in Supplementary Fig 2b were acquired using a Plan-Apochromat ×40/1.0 DIC M27 objective on a Zeiss Axio Imager.M2 equipped with a PIXIS 1024 CCD camera (Pronceton Instruments).

**Western blotting**. Worms were lysed in 1× NuPage LDS sample buffer supplemented with 1× NuPAGE Reducing agent (Thermo Fisher Scientific) and heated at 90 °C for 10 min. Protein were resolved on precast NuPAGE Novex 4–12% Bis–Tris gels (Invitrogen). The proteins were transferred on a nylon membrane using the wet transfer Novex Bolt Mini Blot Module (Invitrogen) at 30 V for 2.5 h. The primary antibodies used included anti-CSR-1 (a gift from the Claycomb laboratory, dilution 1:3000) and anti-GAPDH (Ab125247, dilution 1:2000), and the secondary antibodies included anti-rabbit (31469, Pierce) and anti-mouse (31430, Pierce) used at a dilution of 1:10000. The SuperSignal West Pico PLUS Chemiluminescent Substrate was used to detect the signal using a ChemiDoc MP imaging system (Biorad).

**Collection of early embryos populations**. Synchronous populations of worms were grown on NGM plates until adulthood and were carefully monitored using a stereomicroscope and bleached shortly after worms started to produce the first embryos. After bleaching, Early embryos were washed with cold M9 buffer to slow down embryonic development and immediately frozen in dry ice. A small aliquot of embryo pellet (2 µL) was taken right before freezing and mixed with 10 µL VECTASHIELD® Antifade Mounting Medium with DAPI (Vector laboratories) and immediately frozen in dry ice. DAPI-stained embryos were defrosted on ice and used for counting cell nuclei and score the embryonic cell stage of the population.

*Collection of CSR-1-depleted early embryos populations.* Early embryos were collected as described above with the exception that worms were transferred on auxin or ethanol plates during late L4.

*Collection of early- and late CSR-1-depleted embryo populations.* Early populations of CSR-1-depleted embryos were collected as described above with the exception that harvesting was performed at the very beginning of embryo production to further enrich the population in early staged embryos. After bleaching, an aliquot of the same population was allowed to develop further for 1–2 h in M9 containing 500 μM auxin, 0.5% ethanol or 0.5% ethanol. Embryonic cell stage was scored, and harvesting was performed when embryo populations reached the desired developmental stage.

**Small RNA-seq library preparation.** Total RNA from staged embryo preparations with RIN > 9 was used to generate small RNA libraries. The library preparation was performed similarly to that described in ref. [23].

**RNA IP.** A synchronous population of 40,000 worms (48 h after hatching) or a preparation of at least 150,000 early embryos was collected and suspended in extraction buffer (50 mM HEPES pH 7.5, 300 mM NaCl, 5 mM MgCl$_2$, 10% glycerol, 0.25% NP-40, protease inhibitor cocktails (Thermo Scientific), 40 U/mL RiboLock RNase inhibitors (Thermo Scientific)). Samples were crushed in a metal dounce on ice performing at least 40 strokes. Crude protein extracts were centrifuged at $12,000 \times g$ at 4 °C for 10 min. Protein was quantified using Pierce™ 660 nm Protein Assay Reagent (Thermo Scientific) and 1 mg (for adults) or 700 μg (for embryos) of protein extract was used for RNA immunoprecipitation as described in ref. [23] and used for sRNA-seq library preparation.

**GRO-seq on CSR-1-depleted embryos.** Populations containing at least 40,000 CSR-1-depleted early embryos were collected as described above. Early embryos were resuspended in 1.5 mL Nuclei extraction buffer (3 mM CaCl$_2$, 2 mM MgCl$_2$, 10 mM Tris-HCl pH 7.5, 0.25% Np-40, 10% glycerol, protease inhibitors, and RNase inhibitor 4 U/mL) and transferred to a steel dounce and stroked 40 times. The lysate was cleared from cell debris by centrifuging at $100 \times g$ and nuclei pellet at $1000 \times g$ and washed four times with Nuclei extraction buffer. Nuclei were washed once with Freezing buffer (50 mM Tris-HCl pH 8, 5 mM MgCl$_2$, 0.1 mM EDTA) and resuspended in 100 μL Freezing buffer.

*Nuclear Run-On reaction and RNA extraction.* Nuclear Run-On (NRO) reaction was performed by addition of 100 μL NRO 2× buffer (10 mM Tris-HCl, 5 mM MgCl$_2$, 1 mM DTT, 300 mM KCl, 1% Sarkosyl, 0.5 mM ATP, CTP and GTP, and 0.8 U/μL RNase inhibitor) and using 1 mM Bio-11-UTP final concentration and incubated for 5 min at 30 °C. NRO reaction was stopped by the addition of TRIzol LS reagent (Ambion) and RNA extraction was performed following the manufacturer's instructions. Purified RNA was fragmented by the addition of reverse transcriptase buffer and incubated for 7 min at 95 °C.

*Biotin RNA enrichment.* Biotinylated nascent RNAs were bound to 30 μL Dynabeads MyOne Streptavidin C1 (Invitrogen and) and washed three times as described in ref. [52] and purified with TRIZOL reagent.

*RNA 5′-end repair.* 5′-OH of fragmented RNAs were repaired using polynucleotide kinase (Thermo Scientific) following the manufacturer instructions and incubated at 37 °C for 30 min. RNA was purified with phenol:chloroform and precipitated by the addition of three volumes of ethanol, 1/10th volumes of 3 M sodium acetate and 30 μg glycoblue coprecipitant (Ambion).

*Ligation of 3′ and 5′ Adapter Oligos and second and third biotin enrichments.* RNA was ligated to 3′ end adapter using T4 RNA ligase 2 Truncated KQ (home-made) for 16 h at 15 °C. After ligation RNA was purified using solid-phase reversible immobilization beads (SPRI beads) and biotinylated RNA was enriched as described above. After purification RNA was ligated at 5′ end using T4 RNA ligase 1 for 2 h at 25 °C. RNA was purified using SPRI beads and biotinylated RNA was enriched for a third time as described above.

*Reverse transcription and amplification of cDNA libraries.* Purified RNA was reverse transcribed using SuperScript IV Reverse Transcriptase (Thermo Fisher Scientific) following the manufacturer's conditions except that reaction was incubated for 1 h at 50 °C. cDNA was PCR amplified with specific primers using Phusion High fidelity PCR master mix 2× (New England Biolab) for 18–20 cycles and sequenced on the Illumina Next 500 system.

**Degradome-seq.** Degradome sequencing have been performed as described in ref. [29].

**Sorting of *C. elegans* embryos.** Sorting of *C. elegans* embryos was performed as described in ref. [28] with the following modifications: a strain expressing both

mCherry::OMA-1 and PIE-1::GFP was used to collect embryos enriched in different developmental stages (see Supplementary Fig. 4d, f); embryos after bleaching were fixed with 2% formaldehyde in M9 to block cell division. After sorting, embryos were reverse crosslinked in 250 μL RIPA buffer with RNase inhibitors for 30 min at 70 °C and RNA was extracted with TRIzol LS (Ambion) according to the manufacturer's instructions.

**Strand-specific RNA-seq library preparation.** DNase-treated total RNA with RIN > 8 was used to prepare strand-specific RNA libraries. Strand-specific RNA-seq libraries were prepared using the NEBNext® Ultra™ II Directional RNA Library Prep Kit (NEB, E7765) according to the manufacturer's instructions except that the ribosomal depletion was performed as in ref. [23].

**RT-qPCR.** One microgram DNase-treated total RNA was used as a template for cDNA synthesis using random hexamers and M-MLV reverse transcriptase. qPCR reaction was performed using Applied Biosystems Power up SYBR Green PCR Master mix following the manufacturer's instructions and using an Applied Biosystems QuantStudio 3 Real-Time PCR System. Primers used for qPCR are listed in Supplementary Data 4.

**Single-molecule FISH (smFISH).** Single-molecule FISH (smFISH) was performed as described in ref. [53]. Briefly, embryos were harvested by bleaching, immediately resuspended in methanol at −20 °C, freeze cracked in liquid nitrogen for 1 min, and incubated at −20 °C overnight. Embryos were washed once in wash buffer (10% formamide, 2× SSC buffer (Ambion)) and hybridized with the corresponding FLAP containing probes in 100 μL hybridization buffer (10% dextran sulfate, 2 mM vanadyl-ribonucleoside complex, 0.02% RNAse-free BSA, 50 μg *E. coli* tRNA, 2× SSC, 10% formamide) at 30 ˚C overnight. Hybridized embryos were washed twice with wash buffer and once in 2× SSC buffer before imaging. Right before imaging, embryos were resuspended in 100 μL antifade buffer (0.4% glucose, 10 μM Tris-HCl pH 8, 2× SSC) with 1 μL catalase (Sigma-Aldrich) and 1 μL glucose oxidase (3.7 mg/mL, Sigma-Aldrich) and stained with VECTASHIELD® Antifade Mounting Medium with DAPI (Vector laboratories). DAPI staining was used to select embryos at 20-cell stage. Images of the central plane of embryos at desired developmental stages were acquired on a Zeiss LSM 700 confocal microscopy. Oligos used for smFISH of C01G8.1 mRNA target are listed in Supplementary Data 5.

The detection of single mRNA molecules was performed with the open-source Matlab package FISH-quant[54]. Briefly, RNA signal was enhanced by a two-step convolution of the image with Gaussian Kernels. First, the image background obtained by convolution with a large Gaussian Kernel was estimated and then subtracted. Second, the resulting image was filtered with a small Gaussian Kernel to further enhance the signal-to-noise ratio. RNA spots were detected with a local maximum algorithm. For each embryo, a manually drawn outline was used to limit the detection to the somatic blastomeres and exclude the germline blastomere.

**Ribo-seq.** Ribo-seq has been performed as described in ref. [55] with some modifications. Briefly, embryos harvested by bleaching were lysed by freeze grinding in liquid nitrogen in Polysome buffer (20 mM Tris-HCl pH 8, 140 mM KCl, 5 mM MgCl$_2$, 1% Triton X-100, 0.1 mg/mL cycloheximide) and 1 mg extract was digested by RNase I (100 U) at 37 °C for 5 min and then fractionated on sucrose gradient (10–50%) by ultracentrifugation at $187,000 \times g$ in a SW41-Ti rotor (Beckman Coulter). RNA from monosome fraction was DNase treated and fragments of 28–30 nucleotides were size selected after running on a 15% TBE-Urea gel. 28–30 nucleotide ribosome-protected fragments (RPF) were cloned using the sRNA-seq library preparation method except that the 3′phosphate was removed and 5′ end was phosphorylated by treating RNA with polynucleotide kinase.

**Sequencing data analyses.** All the sequencing data were demultiplexed with Illumina bcl2fastq converter (version v2.17.1.14) and a quality control was performed with fastQC (version v0.11.5).

For Deg-seq data, raw reads were trimmed at their low-quality 3′ end using seqtk (version 1.3-r115-dirty) as follows: seqtk trimfq -b 4. Trimmed reads were aligned to the *C. elegans* genome sequence (ce11, *C. elegans* Sequencing Consortium WBcel235) using Bowtie2 (ref. [56]) with default settings. The alignment was also used to generate the bigwig file with bamCoverage from deepTools (version 3.4.3), normalized using counts per million.

For Ribo-seq data, The 3′ adapter was trimmed from raw reads using Cutadapt[57] v.1.18 using the following parameter: -a TGGAATTCTCGGGTGCCAAGG -discard-untrimmed. Trimmed reads of size ranging from 28 to 30 nucleotides were selected using bioawk (https://github.com/lh3/bioawk). The selected 28–30-nucleotide reads were aligned to the *C. elegans* genome sequence (ce11, *C. elegans* Sequencing Consortium WBcel235) using Bowtie (ref. [56]) v.2.3.4.3 with the following parameters: -L 6 -i S,1,0.8 -N 0. Reads mapping on sense orientation on annotated protein-coding genes were considered as RPF. Such reads were extracted from mapping results using samtools 1.9 (ref. [58]) and bedtools v2.27.1 (ref. [59]) and re-mapped on the genome.

Metaprofiles were generated using RPM from sRNA-seq analysis by summarizing normalized coverage information (taken from bigwig files and

averaged across replicates) along early-degraded targets or late degraded targets using deeptools[60]. For RNA-seq, GRO-seq, Degradome-seq, and Ribo-seq, read counts associated to annotated genes were obtained from the alignments using featureCounts v.1.6.3 (ref. [61]). Normalized abundances were estimated from read counts by first normalizing by transcript union exon length (reads per kilobase, RPK), then by million total RPK (transcripts per million, TPM). Mean TPM was computed by taking the mean across replicates. For Degradome-seq data, degradation efficiency was measured as the $\log_2$ of the ratio between mean TPM of Degradome-seq and RNA-seq reads. For Ribo-seq data, translational efficiency was measured as the $\log_2$ of the ratio between mean TPM of ribosomal protected fragments and of transcripts from RNA-seq, adding a pseudo-count of 1 to both terms of the ratio (in order to avoid discarding genes for which no transcript was detected in RNA-seq).

*Calculation of enrichment factor.* Predicted CSR-1 embryonic targets in different gene expression categories have been calculated as follows, considering the total number of *C. elegans* protein-coding genes (20,447): [(total number of CSR-1 embryonic targets) × (number of genes in the gene expression category)/total number of *C. elegans* protein-coding genes].

Enrichment have been calculated as the ration between observed and predicted genes for each gene expression category.

*RSCU calculation.* RSCU for earlier and later degraded mRNAs was calculated using a CAI calculator (http://genomes.urv.es/CAIcal/).

**Gene lists.** Gene lists are provided in Supplementary Data 1.

**Reporting summary.** Further information on research design is available in the Nature Research Reporting Summary linked to this article.

## Data availability

All sequencing data (Degradome-seq, GRO-seq, Ribo-seq, RNA-seq, and sRNA-seq from total lysate or IP experiments) that support the findings of this study have been deposited in Gene Expression Omnibus (GEO) database with the accession code GSE146062. All other data supporting the findings of this study are available from the corresponding author on reasonable request. Source data are provided with this paper.

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

## Acknowledgements

We would like to thank all the members of the Cecere laboratory and Nicola Iovino for the helpful discussions on the paper and suggestions on the project. We thank G. Riddihough from Life Science Editors for comments on the manuscript. We thank the Mello and the Dumont laboratories for sharing strains and reagents. Some strains were provided by the CGC, funded by NIH Office of Research Infrastructure Programs (P40 OD010440). This project has received funding from the Institut Pasteur, the CNRS, and the European Research Council (ERC) under the European Union's Horizon 2020 research and innovation program under grant agreement no. ERC-StG-679243. E.C. was supported by a Pasteur-Roux Fellowship program. P.Q. was supported by Ligue Nationale Contre le Cancer (S-FB19032).

## Author contributions

G.C. and P.Q. identified and developed the core questions addressed in the project. G.C. wrote the paper with the contribution of P.Q., E.C., and M.S. P.Q. performed most of the experiments and analyzed the results together with G.C. E.C. and L.B. generated all the lines used in this study with the help of P.Q. M.S. performed the immunoprecipitation of CSR-1 and small RNA sequencing in adult worms and performed the ribo-seq experiments together with P.Q. B.L. performed all the bioinformatic analysis. F.M. analyzed the smFISH experiments. C.D. helped P.Q. in setting up the embryo FACS sorting.

## Competing interests

The authors declare no competing interests.
