## [Peer Review File · Nature Communications]

REVIEWER COMMENTS

Reviewer #1 (Remarks to the Author):

Quarato and collaborators report here a potential mechanism responsible for the clearing of maternally inherited mRNA during MZT in animals. The authors suggest that this clearing is mediated by the endonuclease or slicing activity of the Argonaute CSR-1 loaded by maternally inherited 22G small RNA in the nematode *C. elegans*.

While most of the experiments presented here support the involvement of CSR-1 in the repression of maternal transcripts in the embryo, the study, as it stands, feels short in demonstrating the actual contribution of CSR-1 slicing activity in the degradation of maternal mRNAs. The fact that animals expressing CSR-1 slicing defective (ADH) phenocopy the embryonic lethality seen in *csr-1* null animals cannot be interpreted as a molecular proof for the involvement of slicer CSR-1 in clearing maternal mRNAs. The following experiments are needed to support their interesting and quite provocative model fully. If the authors can address those issues, I believe that this study can become extremely important for the small RNA field and, more broadly, to understand how mRNA clearance occurs in animals. In this case, this study will become a strong candidate for Nature Communications.

-To fully support the involvement of the slicing activity of CSR-1 in this process, it will be essential to monitor the effect on maternal mRNA levels in animals expressing CSR-1 (ADH) mutant and compare it with the data obtained from *csr-1* null animals (e.g. monitoring the effect on maternal mRNA levels targeted by CSR-1).

-The authors strongly emphasize the importance of the CSR-1 slicing activity for the clearance of maternal mRNA, but no direct evidence found in this manuscript supports their claims. To demonstrate that CSR-1 cleaves mRNA targets, the authors should, at least detect some of those cleavage products and show their absence CSR-1(ADH) animals (by Northern blotting or 3' RACE-sequencing).

Besides those key experiments essential to demonstrate the role of CSR-1 slicing activity in this process, the following also needs to be addressed:

- As shown in Figure 1D, the auxin treated *mosSCI CSR-1wt* line does not seem to rescue fully the embryonic lethality. It is possible that transgenic animals do not have the same level of expression as wild-type animals. Using Western blotting to compare CSR-1 levels in transgenic animals with the ones in wild-type animals will be necessary here to rule out any effect of a decrease of CSR-1 expression (this is even more particularly important as transgene expression in *C. elegans* germline of often getting silenced).

-It will be important to mention that others slicing competent Argonautes do exist in *C. elegans* germline (PRG-1, ALG-1, ALG-2) and elaborate further in the discussion why they believe CSR-1 can be the sole participant in this developmental process.

-In the Discussion Section, it will be essential to mention that Ago3, along with Ago2, has a slicer activity in mammals (Park et al., NAR, 2017).

Reviewer #2 (Remarks to the Author):

In this manuscript, Quarato et al. discover that the Argonaute protein CSR-1 which is loaded with small RNAs made in the maternal germline, is responsible for the decay of maternal mRNAs in somatic cells of the *C. elegans* embryo. The function of CSR-1 has been mostly studied in the germline, where it has been difficult to tease out its contribution to gene regulation. This study provides good explanations for the function of CSR-1 in the embryo. It was also largely unknown how maternal

mRNAs are cleared in *C. elegans*. In other animals this is done by miRNAs, piRNAs or RNA binding proteins, but not in the worm. It is intriguing that in *C. elegans* another Argonaute protein, one with the ability to slice target mRNAs, has taken on this function. Interestingly, the authors propose that maternal mRNA slicing by an Argonaute protein may be a conserved mechanism for maternal mRNA clearance in animals. These findings will be very relevant for the RNA biology community but also for the broader field of developmental biology.

Specifically, the authors show that:

1. In addition to its known localization in the germline, CSR-1 is present in somatic cells in the early embryo. CSR-1 and its slicer activity are required for embryogenesis. The authors generated thoughtful and solid genetic tools to assess CSR-1 function.
2. In the embryo, CSR-1 is loaded with 22G RNAs that target a different set of mRNAs than in the germline. The mRNAs targeted by these 22G RNAs are derepressed in CSR-1 depleted embryos, in a dose-dependent manner. This strongly suggests that CSR-1, guided by 22G RNAs, triggers target degradation.
3. A large fraction of maternal mRNAs that are cleared in the embryo are targets of CSR-1 and depletion of CSR-1 results in an overall increase in mRNAs that should be cleared in the embryo. Maternal mRNA clearance by CSR-1 happens in the early embryo. The slicer activity of CSR-1 is required.
4. Among the maternally-degraded mRNAs some are degraded earlier than others. The authors find that the early-degraded mRNAs correlate with a higher level of targeting 22G RNAs over the coding sequence, and lower ribosome occupancy.
5. To test whether ribosome occupancy affects maternal mRNA clearance by CSR-1 the authors generated two transgenes that differ only in their 3'UTRs. The authors picked a 3'UTR from a maternal mRNA that is degraded early in the embryo, and one that is degraded late. The 3'UTR can affect different aspects of the life of an mRNA, including translational efficiency. They find that the transgene with the 3'UTR associated with early degradation has decreased levels in embryos, lower ribosome density and higher 22G RNA density. Supporting the idea that low translational efficiency is associated with CSR-1 mediated mRNA clearance

Overall, the experiments are of high quality and support the author's conclusions. There is one point though, that the authors may want to interpret more cautiously: in the last experiment with the different 3'UTRs, the authors infer that they are manipulating translational efficiency and that has an impact on 22G RNA density and rate of clearance in the embryo. However, the direction of causality is not unambiguous in my opinion. Changing the 3'UTR could in principle determine the density of 22G RNAs and this could in turn affect clearance and translational efficiency. I fail to see how this experiment teases out causality rather than further supporting the correlation between ribosome occupancy and 22G RNA density, but perhaps the authors can provide a better explanation for this. In any case, I don't think this diminishes the importance or interest in the work and this particular set of observations, but the authors might want to leave the interpretation a bit more open.

Other minor points:

1. There are a few typos and grammatical errors in the introduction and an unformatted reference.
2. The order of extended data 2 and 3 should be exchanged to match the order of appearance in the text
3. In figure 1 and associated text and suppl. figure, the auxin treatment "from oogenesis" is confusing because it gives the impression that the authors looked at the animals they treated since they were oocytes (as compared to from L1 larvae). I would recommend calling it "from late L4".
4. Fig 2C, D, the bins represent density of targeting 22G RNAs as far as I understand, but this is not clear from the graph label (CSR-1 targets (RPM) gives a very different idea of what is plotted) and is not accurately explained in the legend.
5. The embryonic CSR-1 targets are more than the maternal cleared mRNAs (Fig 3). It would be important to describe what are the other targets of CSR-1. I am not suggesting any additional

experiments, only looking at the available data to see if the other targets of CSR-1 have anything in common.

6. In Fig 3C the authors show for two embryonic CSR-1 targets that they require CSR-1 catalytic activity for their normal clearance. How did the authors select the targets? Did they look at any other targets? This should be stated in the text, figure legend or methods.

7. For the experiment in Figure 5, how did the authors select the two 3'UTRs they use? Are these the only ones they tried? This should be stated in the text, figure legend or methods.

8. In Figure 5b the authors want to show the relative abundance of the two transgenes with the different 3'UTRs at different timepoints. Showing the ratios between the two is however not very informative. It would be best to show the individual samples in a plot in the style of 4A.

Reviewer #3 (Remarks to the Author):

In this manuscript, Quarato et. al., investigated the role of small RNAs in clearance of maternal mRNAs in *C. elegans*. Using an elegant combination of genetics, cell sorting, and deep sequencing approaches, the authors described that clearance of hundreds of maternal mRNAs involves CSR-1 Argonautes and its catalytic activity. They further made an interesting observation that translation status of the embryonic mRNAs exhibit different dynamics of mRNA clearance and small RNA levels: early-cleared maternal mRNAs are less engaged with ribosomes than late-degraded mRNAs and exhibit a different pattern of CSR-1 associated 22G-RNAs. Overall the link between small RNA, mRNA clearance, and translation should be of interest to the broad readership of Nature Communications. However, some conclusions described by the authors are not yet supported by the presented data and additional analyses and experiments are needed. In addition, given the various roles of CSR-1 proposed in previous studies, the authors need to offer explanations in cases where conflicts exist and determine whether CSR-1-mediated mRNA decay is restricted to somatic blastomeres or a more global mechanism in regulating the levels of germline transcripts. If so, the authors should adjust their model accordingly.

Major points:

1. The authors have used both cell biology and deep sequencing approaches to identify the clear role of CSR-1 in degradation of hundreds of maternal mRNAs in embryos. Throughout the manuscript the authors describe the role of CSR-1 in mRNA clearance in somatic blastomeres. The main model proposed by the authors is that CSR-1 specifically clears mRNAs that are no longer engaged in translation in somatic blastomeres. However, it is unclear whether CSR-1-dependent mRNA clearance can only occur in somatic blastomeres of embryos, or in both somatic and germline blastomeres. In addition, whether such mRNA repression occurs only in embryos, or does CSR-1 repress targets in the adult germline? As several studies have identified various roles of CSR-1 in the regulation of germline and embryo mRNA levels, it is important for the authors to compare the data from these studies and explore the possibility of alternative models (also see comments below).

2. One previous study (Gerson-Gurwitz et. al., Cell 2016) identified that CSR-1 and its slicing activity is required to repress (tune) some germline transcripts, resulting in the abnormal level of CSR-1 target genes in early embryos (as early as 1-cell embryos). Therefore, it seems that CSR-1 can repress mRNA levels in the germline. The author should specifically examine this possibility. For example, in Figure 3D the authors intentionally exclude the measurements for the level of germline transcripts in the germline blastomere. In addition, the authors describe that the role of CSR-1 in mRNA clearance does not conflict with the previously proposed role of CSR-1 in licensing germline transcripts, but do the levels of CSR-1 targets increase or decrease in the adult following CSR-1 depletion?

3. The authors made an interesting observation that CSR-1 targets different sets of genes in the adult compared to embryos. However, no specific explanation was provided to address why there are such

differences. The authors should explore whether the difference is caused by differences in target gene mRNA levels, since the mRNAs serve as the template for 22G-RNA production. For example, it is possible that male-specific genes may be present at higher levels in young adults than in embryos, thus contributing to this difference in CSR-1 target genes in adults compared to embryos. Moreover, considering the model proposed by the authors that translation inhibits CSR-1 22G-RNA synthesis within the CDS of mRNAs in the embryos, you would expect the difference can be partially explained by the translation state of various genes in the germline and in the embryos. For example, can a similar trend be observed in germline genes, where constitutively translated genes have more 3' bias of CSR-1 22G-RNA distributions compared to genes that are only translated in specific stages of the germline (such as oogenesis-specific *oma-1*)?

4. Another previous study (Fassnacht et. al., Plos Genetics 2018) has identified the role of CSR-1 in preventing the activation of embryonic transcription of a group of early embryonic genes. However, in this study the authors did not observe changes in transcription after CSR-1 depletion in early embryos. Currently, the authors do not offer explanations for such differences.

5. The observation that translational status of the mRNA correlates with degradation timing and distribution of CSR-1 22G-RNAs is quite interesting. The authors propose a model that ribosome/translation may inhibit the synthesis of CSR-1 22G-RNA and prevent the RNA from CSR-1 slicing. The reporters with different 3' UTRs offer a great way to further investigate the relationship of these processes. However, while the level of small RNAs is higher within the mCherry region of the early-degraded (*egg-6*) compared to the late-degraded (*tbb-2*) 3' UTR, the distribution did not reflect the global analyses shown in Fig 4C. More specifically, a significant and similar amount of 22G-RNAs are made at the 5' end of mCherry in the late-degraded mRNA (*tbb-2*). The authors did not offer explanations for this. Is it possible the 22G-RNAs made at the 5' end of the mCherry gene were not loaded to CSR-1? The CSR-1 IP small RNA cloning experiments in these strains may provide important insights.

In addition, these reporters offer great opportunities to strengthen the function of CSR-1 mRNA decay in embryos and address where these processes occur. However, currently the authors have only shown the behavior of these reporters in wild type animals. The authors should include analysis of these reporters in the background of CSR-1 depletion, as well as rescue with wild type (DDH) CSR-1 and catalytic dead (ADH) CSR-1 to show that mRNA and small RNA levels of early-degraded/ late-degraded reporter constructs are dependent on CSR-1. Additional mCherry FISH analyses on the mCherry reporter mRNAs in wild type and CSR-1 mutant would be helpful to determine whether the CSR-1-mediated mRNA clearance only occurs in somatic blastomeres.

Minor points:

1. As MosSci transgenes are prone to various levels of silencing among strains, the authors need to show the catalytic dead CSR-1 mutant is expressed at the same levels as WT (ADH vs. DDH).

2. It is known that FLAG immunostaining tends to have a high background. The authors should also include mCherry images of the CSR-1 transgene to show localization of CSR-1 in both germline and somatic blastomeres of early embryos.

3. Figure 2A shows separate categories of CSR-1 targeted genes based on expression level. According to the labels, it seems that these categories are not exclusive, as the >1 RPM category should contain the targets in the >50 RPM and >150 RPM categories. However, in Figure 2D, the RNA-seq data corresponding to these categories shows exclusivity, as some data points (such as the highest data point) contained in the >50 and >150 RPM categories are not found in the >1 RPM category. This should be clarified in the figure legend if, for example, outliers have been excluded in the boxplots leading to this discrepancy.

4. Representative FISH images should be shown in Figure 3, not just in supplemental figures.

5. It is not clear why the number of maternally degraded mRNAs (1320) are fewer than the

combination of the number of early-degraded mRNA (482) and late-degraded mRNAs (1572). It is also unclear if the late-degraded mRNAs are also cleared by the CSR-1 pathway in later embryos. In supplementary information, the authors should provide these groups of genes and their normalized expression levels at different embryo stages shown in Figure 3A and Figure 4A. In addition, the levels of CSR-1 22G-RNAs in different stages of the worms (adults, 1-cell, early embryos and late embryos) should also be provided as supplemental information.

6. Text description and figures shown in the manuscript need to be more specific, rather than generalizing the whole groups of genes. In Figure 3C, since only two mRNAs are examined, authors should describe that the two selected genes are increased instead of simply generalizing to maternal cleared mRNAs. Similarly, In Figure 3D, the specific name of the transcript (C01G8.7) should appear in the figure, instead of labeling the figure: maternal cleared CSR-1 target. In addition, the authors choose different genes to examine in various assays. For example, they choose *cpg-1* 22G-RNA levels in Figure 2B, and examine two other genes in the qRT-PCR assay in Figure 3C, along with another gene C01G8.7 in Figure 3D. To avoid the impression of "cherry-picking" the specific data that fit their model, the authors should either be consistent with genes being examined or offer explanations for why different genes are chosen in different assays.

7. Since the authors have not yet shown that translation inhibits CSR-1 function or the RDRP accessibility, question marks should be provided in the model, Figure 6, to indicate this.

8. The following changes are suggested for the descriptions. On page 5, the lead sentence for the first section of the results should be changed to "CSR-1 localized to both somatic and germline blastomeres and its slicer activity is essential for embryonic development". On page 9, the author should change the description of the CSR-1 loaded 22G-RNAs of early-degraded mRNA to reflect the CSR-1-loaded 22G-RNAs are still enriched at the 3' end, but also produced along the whole body of the genes. On page 10, the last sentence of the first paragraph about the "accessibility of 22G-RNA cleavage by CSR-1" seems incorrect. Those observations only "imply ribosome could influence the accessibility of the template for 22G-RNA synthesis".

REVIEWER COMMENTS

Reviewer #1 (Remarks to the Author):

Quarato and collaborators report here a potential mechanism responsible for the clearing of maternally inherited mRNA during MZT in animals. The authors suggest that this clearing is mediated by the endonuclease or slicing activity of the Argonaute CSR-1 loaded by maternally inherited 22G small RNA in the nematode *C. elegans*.

While most of the experiments presented here support the involvement of CSR-1 in the repression of maternal transcripts in the embryo, the study, as it stands, feels short in demonstrating the actual contribution of CSR-1 slicing activity in the degradation of maternal mRNAs. The fact that animals expressing CSR-1 slicing defective (ADH) phenocopy the embryonic lethality seen in *csr-1* null animals cannot be interpreted as a molecular proof for the involvement of slicer CSR-1 in clearing maternal mRNAs. The following experiments are needed to support their interesting and quite provocative model fully. If the authors can address those issues, I believe that this study can become extremely important for the small RNA field and, more broadly, to understand how mRNA clearance occurs in animals. In this case, this study will become a strong candidate for Nature Communications.

We would like to thank the Reviewer #1 for the thoughtful and thorough review of our manuscript and for the constructive comments and rigorous suggestions, which helped to solidify the role of slicing activity of CSR-1 in clearing maternal mRNAs in embryos. Please find below our responses to specific comments.

-To fully support the involvement of the slicing activity of CSR-1 in this process, it will be essential to monitor the effect on maternal mRNA levels in animals expressing CSR-1 (ADH) mutant and compare it with the data obtained from *csr-1* null animals (e.g. monitoring the effect on maternal mRNA levels targeted by CSR-1).

*We agree with the Reviewer #1 that it would be desirable to measure the level of CSR-1 embryonic mRNA targets in absence of CSR-1 catalytic activity to fully prove the involvement of CSR-1 slicer activity on maternal mRNAs. Unfortunately, *csr-1* catalytic mutation also causes strong germline defects and sterility (Extended Data Fig. 2), and therefore cannot be used to detect mRNAs in embryos. For this reason, we have performed RNA-seq experiments in embryos depleted of endogenous CSR-1 and complemented with transgenic expression of CSR-1 DDH (WT) or CSR-1 ADH (Catalytic mutant) proteins. Our results, displayed in Fig. 3e, show an upregulation of CSR-1 targets in animals expressing single-copy transgenic CSR-1 ADH compared to CSR-1 DDH, suggesting the requirement of CSR-1 catalytic activity to clear maternal mRNAs. Also, the expression of the two transgenic proteins is similar (Extended Data Fig. 3c), which allowed us to compare the two transgenic strains.*

-The authors strongly emphasize the importance of the CSR-1 slicing activity for the clearance of maternal mRNA, but no direct evidence found in this manuscript supports their claims. To demonstrate that CSR-1 cleaves mRNA targets, the authors should, at least detect some of those cleavage products and show their absence CSR-1(ADH) animals (by Northern blotting or 3' RACE-sequencing).

We agree with the Reviewer #1 that the detection of CSR-1 cleaved products would be nice to show. However, the small RNAs loaded into CSR-1 protein span the whole region of CSR-1 mRNA targets and is therefore difficult to detect a specific intermediate cleavage product. For this reason, we have performed degradome sequencing method, which allows to clone 5' monophosphate mRNA fragments derived from Argonaute slicing activity, to measure the degradation efficiency (degradome-seq / RNA-seq ratio) of CSR-1 targets in presence or absence of CSR-1 slicer activity. Our results, presented in Fig. 3g, show reduced degradation efficiency of upregulated CSR-1 embryonic targets in animals depleted of CSR-1 and complemented with transgenic expression of CSR-1 ADH compared to CSR-1 DDH. Therefore, the upregulation of CSR-1 target observed in ADH mutant (Fig. 3e) results from impaired degradation of its target mRNAs. We think that these new data (Fig. 3e, g) together with the in vitro evidences of CSR-1 cleavage activity on targeted mRNAs (Aoki et al., 2007) strongly support the conclusion that CSR-1 cleaves target transcripts.

Besides those key experiments essential to demonstrate the role of CSR-1 slicing activity in this process, the following also needs to be addressed:

- As shown in Figure 1D, the auxin treated *mosSCI* CSR-1wt line does not seem to rescue fully the embryonic lethality. It is possible that transgenic animals do not have the same level of expression as wild-type animals. Using Western blotting to compare CSR-1 levels in transgenic animals with the ones in wild-type animals will be necessary here to rule out any effect of a decrease of CSR-1 expression (this is even more particularly important as transgene expression in *C. elegans* germline is often getting silenced).

*We agree with the Reviewer#1 that the auxin treated *MosSCI* CSR-1 DDH is not fully WT. Indeed, our brood size assay shown in Fig. 1d revealed the presence of some animals with high percentage of embryonic lethality*

indicating that the transgenic expression is not sufficient to fully rescue the CSR-1 depletion. To avoid selecting silenced lines, we have checked the level of GFP in the rescued CSR-1 DDH (and ADH) and excluded animals lacking GFP expression in all our broodsize experiments (we have explained this in the method section). However, even after excluding silenced embryos we still observed animals with some percentage of embryonic lethality (Fig. 1d). This suggests that the transgenic protein might not be expressed at the same level as the wild type protein. To verify the levels of transgenic CSR-1 proteins in whole embryo extracts, we have performed as suggested by the Reviewer #1 western blotting analysis of embryos expressing wild-type CSR-1 protein, CSR-1 depleted protein, and CSR-1 depleted protein complemented with transgenic expression of CSR-1 DDH or CSR-1 ADH. The results are shown in Extended Fig. 3c and show a reduced expression of transgenic CSR-1 (DDH and ADH) proteins compared to the endogenous CSR-1. Importantly, the transgenic CSR-1 DDH and CSR-1 ADH are expressed at similar levels, but only the CSR-1 ADH strain show 100% embryonic lethality. Therefore, even if the transgenic CSR-1 DDH does not fully rescue the lack of endogenous, CSR-1 can be still used to compare the effect of CSR-1 ADH vs. CSR-1 DDH. This also explains the different upregulation levels in Fig. 3b and Fig. 3e.

-It will be important to mention that others slicing competent Argonautes do exist in *C. elegans* germline (PRG-1, ALG-1, ALG-2) and elaborate further in the discussion why they believe CSR-1 can be the sole participant in this developmental process.

We thank the Reviewer #1 for this comment and have further elaborated this in the discussion.

-In the Discussion Section, it will be essential to mention that Ago3, along with Ago2, has a slicer activity in mammals (Park et al., NAR, 2017).

We thank the Reviewer #1 for noticing this and we have mentioned in the discussion the slicer activity of Ago3.

Reviewer #2 (Remarks to the Author):

In this manuscript, Quarato et al. discover that the Argonaute protein CSR-1 which is loaded with small RNAs made in the maternal germline, is responsible for the decay of maternal mRNAs in somatic cells of the *C. elegans* embryo. The function of CSR-1 has been mostly studied in the germline, where it has been difficult to tease out its contribution to gene regulation. This study provides good explanations for the function of CSR-1 in the embryo. It was also largely unknown how maternal mRNAs are cleared in *C. elegans*. In other animals this is done by miRNAs, piRNAs or RNA binding proteins, but not in the worm. It is intriguing that in *C. elegans* another Argonaute protein, one with the ability to slice target mRNAs, has taken on this function. Interestingly, the authors propose that maternal mRNA slicing by an Argonaute protein may be a conserved mechanism for maternal mRNA clearance in animals. These findings will be very relevant for the RNA biology community but also for the broader field of developmental biology.

Specifically, the authors show that:

1. In addition to its known localization in the germline, CSR-1 is present in somatic cells in the early embryo. CSR-1 and its slicer activity are required for embryogenesis. The authors generated thoughtful and solid genetic tools to assess CSR-1 function.
2. In the embryo, CSR-1 is loaded with 22G RNAs that target a different set of mRNAs than in the germline. The mRNAs targeted by these 22G RNAs are derepressed in CSR-1 depleted embryos, in a dose-dependent manner. This strongly suggests that CSR-1, guided by 22G RNAs, triggers target degradation.
3. A large fraction of maternal mRNAs that are cleared in the embryo are targets of CSR-1 and depletion of CSR-1 results in an overall increase in mRNAs that should be cleared in the embryo. Maternal mRNA clearance by CSR-1 happens in the early embryo. The slicer activity of CSR-1 is required.
4. Among the maternally-degraded mRNAs some are degraded earlier than others. The authors find that the early-degraded mRNAs correlate with a higher level of targeting 22G RNAs over the coding sequence, and lower ribosome occupancy.
5. To test whether ribosome occupancy affects maternal mRNA clearance by CSR-1 the authors generated two transgenes that differ only in their 3'UTRs. The authors picked a 3'UTR from a maternal mRNA that is degraded early in the embryo, and one that is degraded late. The 3'UTR can affect different aspects of the life of an mRNA, including translational efficiency. They find that the transgene with the 3'UTR associated with early degradation has decreased levels in embryos, lower ribosome density and higher 22G RNA density. Supporting the idea that low translational efficiency is associated with CSR-1 mediated mRNA clearance.

Overall, the experiments are of high quality and support the author's conclusions. There is one point though, that the authors may want to interpret more cautiously: in the last experiment with the different 3'UTRs, the authors infer that they are manipulating translational efficiency and that has an impact on 22G RNA density and rate of clearance in the embryo. However, the direction of causality is not unambiguous in my opinion. Changing the 3'UTR could in principle determine the density of 22G RNAs and this could in turn affect clearance and translational efficiency. I fail to see how this experiment teases out causality rather than further supporting the correlation between ribosome occupancy and 22G RNA density, but perhaps the authors can provide a better explanation for this. In any case, I don't think this diminishes the importance or interest in the work and this particular set of observations, but the authors might want to leave the interpretation a bit more open.

First, we would like to thank the Reviewer #2 for this comprehensive summary and for appreciating the novelty of our work and the broad implication of our finding for the RNA and Developmental biology communities. In regard to the experiment shown in Fig. 5, we agree with the reviewer that we need to be more cautious in the interpretation of our results. Indeed, we have extended the discussion and we have taken into the account alternative interpretations of these results.

Our interpretation of the results presented in Fig. 5 is justified by the notion that the levels of CSR-1 22G-RNAs on the coding sequences of mRNA targets depend on CSR-1 catalytic activity which is counteracted by the presence of ribosomes. This has been shown in adult worms in our recent work deposited on Biorxiv (Singh at al., biorxiv 2020). For these reasons, we tend to support the idea that the translational efficiency caused by the different 3'UTR might alter the capacity of CSR-1 to produce 22G-RNAs and degrade maternal mRNAs.

*Also, in our manuscript we have shown that late and early degraded targets differ in their ability to produce and load CSR-1 22G-RNAs on the coding sequence, even though the level of CSR-1 22G-RNAs on the 3'UTR is similar (Fig. 4d). This feature is also observed on the two endogenous genes *tbb-2* and *egg-6* (see new Extended Data Fig. 8a) and on the same coding sequence (*mCherry*) of the two different transgenes fused with *tbb-2* or *egg-6* 3'UTR (see new Fig. 5e). Given that the density of CSR-1 22G-RNAs derived from the 3'UTR proximal to the coding sequence is similar for the two classes of genes, we also exclude that the high level of CSR-1-bound 22G-RNAs on *mCherry* fused with *egg-6* 3'UTR is caused by differences in the initial endogenous pool of small RNAs that can trigger increased initiation of RdRP-dependent 22G-RNAs. Nonetheless, we cannot exclude that other unknown features of 3'UTRs might influence the initiation of CSR-1 22G-RNAs on the coding sequences.*

Other minor points:

1. There are a few typos and grammatical errors in the introduction and an unformatted reference.

We thank the Reviewer#2 for noticing this and we have modified the typos and the unformatted reference.

2. The order of extended data 2 and 3 should be exchanged to match the order of appearance in the text

We thank the Reviewer#2 for noticing this and we have corrected it.

3. In figure 1 and associated text and suppl. figure, the auxin treatment "from oogenesis" is confusing because it gives the impression that the authors looked at the animals they treated since they were oocytes (as compared to from L1 larvae). I would recommend calling it "from late L4".

We agree with Reviewer#2 and we have changed it in figures, text, and method.

4. Fig 2C, D, the bins represent density of targeting 22G RNAs as far as I understand, but this is not clear from the graph label (CSR-1 targets (RPM) gives a very different idea of what is plotted) and is not accurately explained in the legend.

We have explained it better in figure legend and graph label.

5. The embryonic CSR-1 targets are more than the maternal cleared mRNAs (Fig 3). It would be important to describe what are the other targets of CSR-1. I am not suggesting any additional experiments, only looking at the available data to see if the other targets of CSR-1 have anything in common.

We have expanded our classification of embryonic mRNAs and we have calculated the enrichment of CSR-1 targets for each of the categories identified in Extended Data Fig. 5.

6. In Fig 3C the authors show for two embryonic CSR-1 targets that they require CSR-1 catalytic activity for their normal clearance. How did the authors select the targets? Did they look at any other targets? This should be stated in the text, figure legend or methods.

We have provided the genome-wide data of the experiment originally reported in Fig 3c (new Fig. 3e, f) and also quantified another target using RT-qPCR (the same used for smFISH) in Fig. 3f.

7. For the experiment in Figure 5, how did the authors select the two 3'UTRs they use? Are these the only ones they tried? This should be stated in the text, figure legend or methods.

*We have mentioned in the text that we have successfully generated only these two transgenic lines. The criterion used to choose the two 3'UTRs is based on the observation that early and late degraded transcripts have different amount of CSR-1 22G-RNAs on their coding sequences, even though their level on the 3'UTR is similar. CSR-1-bounds 22G-RNAs on *tbb-2* and *egg-6* genes followed this criterion (Extended Data Fig. 8a).*

8. In Figure 5b the authors want to show the relative abundance of the two transgenes with the different 3'UTRs at different timepoints. Showing the ratios between the two is however not very informative. It would be best to show the individual samples in a plot in the style of 4A.

We have changed the representation of the data in Fig. 5b, as suggested by the Reviewer#2.

Reviewer #3 (Remarks to the Author):

In this manuscript, Quarato et. al., investigated the role of small RNAs in clearance of maternal mRNAs in *C. elegans*. Using an elegant combination of genetics, cell sorting, and deep sequencing approaches, the authors described that clearance of hundreds of maternal mRNAs involves CSR-1 Argonautes and its catalytic activity. They further made an interesting observation that translation status of the embryonic mRNAs exhibit different dynamics of mRNA clearance and small RNA levels: early-cleared maternal mRNAs are less engaged with ribosomes than late-degraded mRNAs and exhibit a different pattern of CSR-1 associated 22G-RNAs. Overall the link between small RNA, mRNA clearance, and translation should be of interest to the broad readership of Nature Communications. However, some conclusions described by the authors are not yet supported by the presented data and additional analyses and experiments are needed. In addition, given the various roles of CSR-1 proposed in previous studies, the authors need to offer explanations in cases where conflicts exist and determine whether CSR-1-mediated mRNA decay is restricted to somatic blastomeres or a more global mechanism in regulating the levels of germline transcripts. If so, the authors should adjust their model accordingly.

*We thank the Reviewer #3 for appreciating our experimental approaches to investigate the role of small RNAs in clearance of maternal mRNA in *C. elegans*. We also thank the reviewer for the careful and insightful review of our manuscript. In this version of the manuscript we have provided more experiments and comments that address all the reviewers' concerns, strengthening our conclusions. Please find below our responses to specific comments.*

Major points:

1. The authors have used both cell biology and deep sequencing approaches to identify the clear role of CSR-1 in degradation of hundreds of maternal mRNAs in embryos. Throughout the manuscript the authors describe the role of CSR-1 in mRNA clearance in somatic blastomeres. The main model proposed by the authors is that CSR-1 specifically clears mRNAs that are no longer engaged in translation in somatic blastomeres. However, it is unclear whether CSR-1-dependent mRNA clearance can only occur in somatic blastomeres of embryos, or in both somatic and germline blastomeres.

The lack of degradation of maternal mRNAs in germline blastomeres has been already observed some time ago by Geraldine Seydoux (Seydoux and Fire, 1994). However, the reasons why these maternal mRNAs are retained in germline blastomeres during early embryogenesis are still unknown. More recently, the Seydoux Lab have shown that these maternal mRNAs in the germline blastomeres accumulate in phase-separated germ granules where they appeared to be protected from degradation (Lee et al., 2020). We have also observed the retention of maternal mRNAs in germline blastomere with our smFISH experiments (see new Fig. 3d). However, because the signal from single mRNA molecules are aggregated in germ granules it is not possible to quantify individual mRNAs by our smFISH approach as it is performed for somatic blastomeres. Also, based on our recent work (Singh et al., biorxiv 2020), where we show that the slicer activity of CSR-1 is only present in the cytosol and not in germ granules, we tend to exclude that in the embryo CSR-1 can cleave target mRNAs in germ granules. We have included a paragraph in the discussion mentioning the above possibilities.

In addition, whether such mRNA repression occurs only in embryos, or does CSR-1 repress targets in the adult germline? As several studies have identified various roles of CSR-1 in the regulation of germline and embryo mRNA levels, it is important for the authors to compare the data from these studies and explore the possibility of alternative models (also see comments below).

*We agree with the Reviewer #3 that it is important to take into account the slicer activity and/or other functions of CSR-1 in adult germline. However, previously published datasets show somehow conflicting evidences that are difficult to compare with our data in a meaningful way. For instance, multiple studies have used either *csr-1* knockout strains to show that CSR-1 promotes the expression of its targets (anti-silencer) or a transgenic expression of CSR-1 catalytic mutant to show that CSR-1 actually slices its mRNA targets (slicer). In addition, CSR-1 mutants display germline developmental defects, which might skew their genome-wide analyses. To resolve the long-standing paradox of CSR-1 function as anti-silencer or a slicer we have developed a worm sorting strategy coupled with genome-wide methods, which allowed us to obtain a population of knockout and catalytic CSR-1 mutants without developmental defects to precisely measure transcription, mRNA stability, translation, 22G-RNA synthesis, and Argonaute loading. By using this approach, we were able to elucidate catalytic-dependent and -independent functions of CSR-1. These results are part of a whole new work on CSR-1 22G-RNA biogenesis, which we have currently deposited on biorxiv (Singh et al., biorxiv 2020). The results we have obtained in adult worms correlates with the finding described here in the embryos (see also specific comments below).*

2. One previous study (Gerson-Gurwitz et. al., Cell 2016) identified that CSR-1 and its slicing activity is required to repress (tune) some germline transcripts, resulting in the abnormal level of CSR-1 target genes in early embryos (as early as 1-cell embryos). Therefore, it seems that CSR-1 can repress mRNA levels in the germline. The author should specifically examine this possibility. For example, in Figure 3D the authors intentionally exclude the measurements for the level of germline transcripts in the germline blastomere. In addition, the authors describe that the role of CSR-1 in mRNA clearance does not conflict with the previously proposed role of CSR-1 in licensing germline transcripts, but do the levels of CSR-1 targets increase or decrease in the adult following CSR-1 depletion?

We have intentionally excluded the signal from germline blastomere in our smFISH because we cannot count single mRNAs in those granules. In regard to the role of CSR-1 in adults, we have systematically investigated CSR-1 catalytic-dependent and independent functions in adult worms in another work recently deposited on biorxiv (Singh et al., biorxiv 2020). Our main findings in adult worms reveal that the catalytic activity of CSR-1 is primarily needed for the biogenesis of CSR-1 22G-RNAs on their coding target transcripts. Nonetheless, we detected some effect on mRNA level on the most abundant CSR-1 targets as described in (Gerson-Gurwitz et. al., Cell 2016). Importantly, we discovered that CSR-1 cleave and degrade mRNA targets primarily in the cytoplasm and not in the germ granules in adult germlines. We have also discovered that CSR-1 can have some protective function from PIWI/piRNA silencing on specific class of mRNAs and we propose this protection is happening in germ granules. Therefore, we think that CSR-1 in somatic blastomeres, where it is exclusively localized in the cytoplasm, primarily cleaves mRNAs and contributes to the clearance of these mRNA targets. Because of the lack of PIWI protein in somatic blastomeres (Fig. 1a) we excluded any protective function of CSR-1 in somatic blastomeres. We have included a paragraph in the discussion mentioning our findings of CSR-1 in adult worms (Singh et al., biorxiv 2020) and how these results can be integrated in our model of CSR-1 function in the embryo.

3. The authors made an interesting observation that CSR-1 targets different sets of genes in the adult compared to embryos. However, no specific explanation was provided to address why there are such differences. The authors should explore whether the difference is caused by differences in target gene mRNA levels, since the mRNAs serve as the template for 22G-RNA production. For example, it is possible that male-specific genes may be present at higher levels in young adults than in embryos, thus contributing to this difference in CSR-1 target genes in adults compared to embryos. Moreover, considering the model proposed by the authors that translation inhibits CSR-1 22G-RNA synthesis within the CDS of mRNAs in the embryos, you would expect the difference can be partially explained by the translation state of various genes in the germline and in the embryos. For example, can a similar trend be observed in germline genes, where constitutively translated genes have more 3' bias of CSR-1 22G-RNA distributions compared to genes that are only translated in specific stages of the germline (such as oogenesis-specific *oma-1*)?

We agree with the Reviewer #3. We have indeed explored all these possibilities in adult worms in a separate study by Singh et al, biorxiv 2020. There, we have investigated the rules governing germline mRNA targeting by CSR-1 and as suggested by the Reviewer 3, we found that the translational status of an mRNAs more than its level of expression determine the abundance of CSR-1 targets. We found that CSR-1 target genes with abundant 22G-RNAs tend to incorporate unfavorable codons which in turn slows the translation rate facilitating CSR-1 22G-RNA production along the gene body.

4. Another previous study (Fassnacht et. al., Plos Genetics 2018) has identified the role of CSR-1 in preventing the activation of embryonic transcription of a group of early embryonic genes. However, in this study the authors did not observe changes in transcription after CSR-1 depletion in early embryos. Currently, the authors do not offer explanations for such differences.

This is correct. We did not observe any transcriptional changes of the CSR-1 targets in CSR-1 depleted embryo by GRO-seq. These results agreed with the GRO-seq performed in Adults ADH versus WT worms (GRO-seq data in Sing et al, biorxiv 2020). The three genes monitored by Fassnacht et. al., Plos Genetics 2018 belong to the very early transcribed genes in embryos (vet-4, vet-6, pes-10) and do not belong to CSR-1 targets in Adult and early embryos (except pes-10 in embryo that has very low density of CSR-1 22G-RNAs). Therefore, any changes in these genes might be not directly linked to CSR-1 slicer activity. For instance, in our Adult dataset we have sorted mutant animals to avoid severe germline defects. Indeed, these genes do not change by GRO-seq in Adult mutants (Singh et al., Biorxiv 2020) and embryos. Therefore, it is possible that the data collected by Fassnacht et. al. might be affected by germline defects, possibly including lack of chromosome compaction in oocyte due to the role of CSR-1 in histone biogenesis (Avgousti et al., 2012).

5. The observation that translational status of the mRNA correlates with degradation timing and distribution of CSR-1 22G-RNAs is quite interesting. The authors propose a model that ribosome/translation may inhibit the synthesis of CSR-1 22G-RNA and prevent the RNA from CSR-1 slicing. The reporters with different 3' UTRs offer a great way to further investigate the relationship of these processes. However, while the level of small RNAs is higher within the mCherry region of the early-degraded (egg-6) compared to the late-degraded (tbb-2) 3' UTR, the distribution did not reflect the global analyses shown in Fig 4C. More specifically, a significant and similar amount of 22G-RNAs are made at the 5' end of mCherry in the late-degraded mRNA (tbb-2). The authors did not offer explanations for this. Is it possible the 22G-RNAs made at the 5' end of the mCherry gene were not loaded to CSR-1? The CSR-1 IP small RNA cloning experiments in these strains may provide important insights.

We thank the Reviewer #3 for this important observation. Indeed, following the reviewer suggestions we have performed CSR-1 IP and small RNA sequencing in the two transgenic strains (data shown in Fig. 5d, e and Extended Data Fig. 7d). We indeed observed increased CSR-1 loading of 22G-RNAs along the whole region of mCherry sequence fused with egg-6 3'UTR. The new data agree with the global analyses shown in Fig. 4c, d).

In addition, these reporters offer great opportunities to strengthen the function of CSR-1 mRNA decay in embryos and address where these processes occur. However, currently the authors have only shown the behavior of these reporters in wild type animals. The authors should include analysis of these reporters in the background of CSR-1 depletion, as well as rescue with wild type (DDH) CSR-1 and catalytic dead (ADH) CSR-1 to show that mRNA and small RNA levels of early-degraded/ late-degraded reporter constructs are dependent on CSR-1. Additional mCherry FISH analyses on the mCherry reporter mRNAs in wild type and CSR-1 mutant would be helpful to determine whether the CSR-1-mediated mRNA clearance only occurs in somatic blastomeres.

We agree with the Reviewer #3 that it would be great to follow the two 3'UTR reporter strains in CSR-1 mutants. However, we cannot cross the two single-copy transgenic 3'UTR reporters with our degron strain, since the same MosSCI site is occupied by the TIR-1 enzyme used for the depletion. The only experiment feasible would be RNAi of CSR-1 at some point of Adult development. However, due to the variable effects of RNAi in each worm it would be difficult to assess the specificity of our assay. Lastly, the smFISH probes designed on the mCherry sequence used for our transgenes did not allowed us to detect clear signal.

Minor points:

1. As MosSci transgenes are prone to various levels of silencing among strains, the authors need to show the catalytic dead CSR-1 mutant is expressed at the same levels as WT (ADH vs. DDH).

We have provided a western blotting in Extended Data Fig. 3c where we have compared CSR-1 DDH and CSR-1 ADH transgenic proteins. They indeed have similar levels.

2. It is known that FLAG immunostaining tends to have a high background. The authors should also include mCherry images of the CSR-1 transgene to show localization of CSR-1 in both germline and somatic blastomeres of early embryos.

We have included the localization of mCherry::CSR-1 in 4-cells embryo by live fluorescent imaging.

3. Figure 2A shows separate categories of CSR-1 targeted genes based on expression level. According to the labels, it seems that these categories are not exclusive, as the >1 RPM category should contain the targets in the >50 RPM and >150 RPM categories. However, in Figure 2D, the RNA-seq data corresponding to these categories shows exclusivity, as some data points (such as the highest data point) contained in the >50 and >150 RPM categories are not found in the >1 RPM category. This should be clarified in the figure legend if, for example, outliers have been excluded in the boxplots leading to this discrepancy.

We thank the Reviewer #3 for noticing this. It was mislabeled in the figure legend and we have clarified it.

4. Representative FISH images should be shown in Figure 3, not just in supplemental figures.

We included representative smFISH images in Fig. 3d.

5. It is not clear why the number of maternally degraded mRNAs (1320) are fewer than the combination of the number of early-degraded mRNA (482) and late-degraded mRNAs (1572). It is also unclear if the late-degraded mRNAs are also cleared by the CSR-1 pathway in later embryos. In supplementary information, the authors should provide these groups of genes and their normalized expression levels at different embryo stages shown in Figure 3A and Figure 4A. In addition, the levels of CSR-1 22G-RNAs in different stages of the worms (adults, 1-cell, early embryos and late embryos) should also be provided as supplemental information.

The late degraded mRNAs are a separate category from the maternal cleared mRNAs. This is because of the threshold used to calculate them. The late degraded transcripts in fact do not change between early embryos and 1-cell embryos and this is the reason why they are not included in the maternal cleared mRNAs, which are lower at early and late stage embryos compared to 1-cell embryos. The early and late degraded mRNAs are therefore categories of genes separate from the maternal cleared mRNAs, but are included in the category of maternal mRNAs. We have further clarified this in the text. We have also included, as requested by the Reviewer #3, the normalized expression levels of all the categories of genes defined in this study and the level of CSR-1-bound 22G-RNAs in Embryos and Adult worms in the new supplementary table 1c.

6. Text description and figures shown in the manuscript need to be more specific, rather than generalizing the whole groups of genes. In Figure 3C, since only two mRNAs are examined, authors should describe that the two selected genes are increased instead of simply generalizing to maternal cleared mRNAs. Similarly, In Figure 3D, the specific name of the transcript (C01G8.7) should appear in the figure, instead of labeling the figure: maternal cleared CSR-1 target. In addition, the authors choose different genes to examine in various assays. For example, they choose cpq-1 22G-RNA levels in Figure 2B, and examine two other genes in the qRT-PCR assay in Figure 3C, along with another gene C01G8.7 in Figure 3D. To avoid the impression of "cherry-picking" the specific data that fit their model, the authors should either be consistent with genes being examined or offer explanations for why different genes are chosen in different assays.

We have now included the C01G8.7 gene in the RT-qPCR assay. Unfortunately, not all the genes were suitable to design smFISH probes. We have also extended our analysis on CSR-1 ADH vs CSR-1 DDH genome-wide by RNA-seq in Fig. 3e.

7. Since the authors have not yet shown that translation inhibits CSR-1 function or the RDRP accessibility, question marks should be provided in the model, Figure 6, to indicate this.

We have done this.

8. The following changes are suggested for the descriptions. On page 5, the lead sentence for the first section of the results should be changed to "CSR-1 localized to both somatic and germline blastomeres and its slicer activity is essential for embryonic development". On page 9, the author should change the description of the CSR-1 loaded 22G-RNAs of early-degraded mRNA to reflect the CSR-1-loaded 22G-RNAs are still enriched at the 3' end, but also produced along the whole body of the genes. On page 10, the last sentence of the first paragraph about the "accessibility of 22G-RNA cleavage by CSR-1" seems incorrect. Those observations only "imply ribosome could influence the accessibility of the template for 22G-RNA synthesis".

We have done this.

REVIEWER COMMENTS

Reviewer #1 (Remarks to the Author):

The authors did an excellent job of answering my comments adequately with this revised manuscript. I am therefore supportive of the publication of this exciting study in Nature Communications after those two following modifications:

1- In the Discussion Section (p.16), it should mention that PRG-1 has a slicing activity. In this paragraph, it will be relevant as well, referring to studies demonstrating the slicing activities of ALG-1/2 and PRG-1.

2- In Extended Figure 3c, please correct the lines numbering mentioned in the Figure legend. They are currently not appropriate.

Reviewer #2 (Remarks to the Author):

The authors have addressed all my comments and concerns, and they have added a substantial amount of new data that further support their conclusions. Specifically, they added a transcriptome-wide analysis of embryos expressing wt or catalytic-dead CSR-1, and degradome sequencing, which both support the role of CSR-1 mediated cleavage in maternal mRNA decay. They also performed CSR-1 immunoprecipitation and analyzed the loaded 22G RNAs and provided additional controls such as western blots that give additional confidence to the comparisons across strains. Overall, I recommend this manuscript for publication.

Reviewer #3 (Remarks to the Author):

In the revised manuscript, Quarato et al. has provided additional data to clarify the function of CSR-1 small RNAs in mRNA clearance in the embryo of *C. elegans*. Overall, this is an interesting paper that links the function of small RNAs with the translational status of maternal mRNAs and their degradation. However, the manuscript have shared many key conclusions with the authors' another manuscript (Singh et. al., bioRxiv 2020), except the experiments are performed at different developmental stages (adult worms vs. embryos). While these results are important and interesting, I am not convinced they warrant two independent manuscripts. In addition, the authors have overstated some of their observations and have yet to address some concerns raised in previous reviewer comments. All these issues need to be properly addressed by re-writing and changing the presentation of existing data. The authors should both soften their various descriptions of CSR-1 function in mRNA degradation/clearance (see details below), and show degradation kinetics of maternal mRNAs in CSR-1 mutants so the readers can evaluate the overall contribution of CSR-1 function in mRNA clearance.

Specifically,

(1) The authors describe that CSR-1 fine-tunes germline mRNA expression in the adult, but clears mRNAs in the embryo. This description strongly implies that CSR-1 degrades targets more effectively or broadly in the embryo compared to in the adult germline. However, results from the authors' other manuscript (Singh et al., bioRxiv 2020) show little difference between the germline and embryo when comparing the amplitude of mRNA upregulation in CSR-1 mutants, as well as the number of genes targeted by CSR-1 (Fig 2d of this paper and Fig 1d from Singh et al., bioRxiv 2020). Their results suggest CSR-1 fine-tunes germline and embryo mRNAs. Therefore, the authors should remove or soften their language when suggesting distinct functions of CSR-1 in the embryo and germline.

(2) The authors describe CSR-1 degrades mRNAs in the somatic but not germline blastomeres without evidence to support this claim. The RNA FISH data in updated Fig 3d indicate mRNA levels are increased "both" in the germline and somatic blastomeres. In addition, their other experiments isolate all embryo RNAs, and therefore do not distinguish whether CSR-1 modulates gene expression only in the somatic and not germline blastomeres. The authors should describe their results as "the reduction of mRNA degradation in the CSR-1 mutant embryo". Potential differences of CSR-1 function in somatic and germline blastomeres should be addressed only in the discussion.

(3) The authors have not provided data on the quantitative contribution of CSR-1 to mRNA clearance. For example, the authors should show the rate of mRNA degradation in CSR-1 mutants, specifically, in Fig 3a for maternal cleared mRNAs or CSR-1 embryonic mRNA targets, and in Fig 4a for early- or late-degraded mRNAs. These data have already been collected for data shown in Fig 3b. This is a critical point of the paper, and currently unclear whether a small group of maternal mRNAs are degraded mainly by CSR-1 or if CSR-1 is merely one mechanism that participates in clearance of these mRNAs. It seems CSR-1 is only one of the mechanisms for mRNA clearance, as even the highest CSR-1 targeted genes (~115 genes with 22G RNA density >150RPM) on average show less than 2 fold mRNA upregulation in CSR-1 mutant early embryos. Supporting this point, the authors describe a CSR-1 independent mechanism for clearance of late-degraded mRNAs. All this considered, the authors should soften their language in the title/manuscript to for example, "a small RNA pathway that facilitates the clearance of untranslated mRNAs".

(4) In the title, abstract, and manuscript line 89 (and some other places), the authors state germline inherited small RNAs are important for clearance/ degradation of maternal mRNAs. However, the authors have not provided experimental evidence showing these small RNAs (CSR-1 22G-RNAs) are indeed inherited from the maternal germline. Instead, their data argue the small RNA population is correlated with the translational status of the mRNAs, raising the possibility some of these small RNAs are made de novo in the embryo. The authors should remove "inherited small RNAs" unless new experiments are provided.

REVIEWER COMMENTS

Reviewer #1 (Remarks to the Author):

The authors did an excellent job of answering my comments adequately with this revised manuscript. I am therefore supportive of the publication of this exciting study in Nature Communications after those two following modifications:

We thank the Reviewer #1 for appreciating our efforts in addressing all the previous concerns.

1- In the Discussion Section (p.16), it should mention that PRG-1 has a slicing activity. In this paragraph, it will be relevant as well, referring to studies demonstrating the slicing activities of ALG-1/2 and PRG-1.

We have included this comment in the text.

2- In Extended Figure 3c, please correct the lines numbering mentioned in the Figure legend. They are currently not appropriate.

We have corrected the lane numbers in the legend of Extended Data Figure 3c.

Reviewer #2 (Remarks to the Author):

The authors have addressed all my comments and concerns, and they have added a substantial amount of new data that further support their conclusions. Specifically, they added a transcriptome-wide analysis of embryos expressing wt or catalytic-dead CSR-1, and degradome sequencing, which both support the role of CSR-1 mediated cleavage in maternal mRNA decay. They also performed CSR-1 immunoprecipitation and analyzed the loaded 22G RNAs and provided additional controls such as western blots that give additional confidence to the comparisons across strains.

Overall, I recommend this manuscript for publication.

We thank the Reviewer #2 for appreciating the additional experiments shown to address all the previous concerns.

Reviewer #3 (Remarks to the Author):

In the revised manuscript, Quarato et al. has provided additional data to clarify the function of CSR-1 small RNAs in mRNA clearance in the embryo of *C. elegans*. Overall, this is an interesting paper that links the function of small RNAs with the translational status of maternal mRNAs and their degradation. However, the manuscript have shared many key conclusions with the authors' another manuscript (Singh et. al., bioRxiv 2020), except the experiments are performed at different developmental stages (adult worms vs. embryos). While these results are important and interesting, I am not convinced they warrant two independent manuscripts. In addition, the authors have overstated some of their observations and have yet to address some concerns raised in previous reviewer comments. All these issues need to be properly addressed by re-writing and changing the presentation of existing data. The authors should both soften their various descriptions of CSR-1 function in mRNA degradation/clearance (see details below), and show degradation kinetics of maternal mRNAs in CSR-1 mutants so the readers can evaluate the overall contribution of CSR-1 function in mRNA clearance.

We thank the Reviewer #3 for appreciating the additional experiments shown to address most of the reviewers' concerns and for suggesting to present our results in a more meaningful way. In this second revised manuscript version we have included new figures and experiments addressing all the concerns raised by the Reviewer #3, and where needed we have rephrased the parts of the manuscript following the reviewer's suggestions.

In regard to the data on mRNA regulation by CSR-1 in adult germlines that we have mentioned in the manuscript by Singh et al., these have been used to present only one main figure (Figure 1, BioRxiv 2020) of the manuscript. These experiments have been performed to resolve the current controversy between the role of CSR-1 protein vs CSR-1 catalytic activity in gene regulation by using *ad hoc* mutants and experimental strategies. However, the main key findings of the manuscript address the biogenesis of CSR-1 22G-RNAs in animal germlines, which is still largely unknown and is the main focus of the manuscript by Singh et al. We don't think that the Figure 1 by Singh et al. will be more appropriately presented in the current manuscript by Quarato et al. given its focus on the gene regulatory role of CSR-1 in embryos. We do believe that the two manuscripts address two important and different biological questions of the CSR-1 small RNA pathway.

Specifically,

(1) The authors describe that CSR-1 fine-tunes germline mRNA expression in the adult, but clears mRNAs in the embryo. This description strongly implies that CSR-1 degrades targets more effectively or broadly in the embryo compared to in the adult germline. However, results from the authors' other manuscript (Singh et al., bioRxiv 2020) show little difference between the germline and embryo when comparing the amplitude of mRNA upregulation in CSR-1 mutants, as well as the number of genes targeted by CSR-1 (Fig 2d of this paper and Fig 1d from Singh et al., bioRxiv 2020). Their results suggest CSR-1 fine-tunes germline and embryo mRNAs. Therefore, the authors should remove or soften their language when suggesting distinct functions of CSR-1 in the embryo and germline.

We agree with the Reviewer #3 with the comments about the fine-tuning vs the clearance and we have clarified this point in the discussion. We do think that the catalytic efficiency of CSR-1 protein is similar in adults and embryos. Our data in both adults and embryos show that CSR-1 is capable of cleaving mRNA targets exclusively at the post-transcriptional level and its effect is dependent on the density of 22G-RNAs antisense to the target mRNA. However, we speculate that the continuous transcription of CSR-1 targets in adult germlines allows CSR-1 to only fine-tune the levels of its targets, whereas the contribution of CSR-1 activity on maternal mRNA targets in early embryos results in mRNA clearance because of the lack of transcription in the early phase of embryogenesis.

(2) The authors describe CSR-1 degrades mRNAs in the somatic but not germline blastomeres without evidence to support this claim. The RNA FISH data in updated Fig 3d indicate mRNA levels are increased "both" in the germline and somatic blastomeres. In addition, their other experiments isolate all embryo RNAs, and therefore do not distinguish whether CSR-1 modulates gene expression only in the somatic and not germline blastomeres. The authors should describe their results as "the reduction of mRNA degradation in the CSR-1 mutant embryo". Potential differences of CSR-1 function in somatic and germline blastomeres should be addressed only in the discussion.

We agree with the Reviewer #3 that we have not provided data showing that CSR-1 exclusively cleaves maternal mRNAs in somatic blastomeres. We agree that by visually inspecting the image in Figure 3d it appears that also the germline blastomere retain more RNA in the germ granules in CSR-1 depleted embryos. However, as explained in the previous rebuttal, we cannot perform a proper quantification of the mRNA in the germline blastomeres. We have clarified in the text the experiments showing the reduction of mRNA degradation in *csr-1* mutant using whole embryos. However, in our smFISH experiments we still think is important to point out that CSR-1 also degrades mRNAs in somatic blastomeres because these are the cells where the clearance of maternal mRNAs has been documented. Therefore, the unexpected accumulation of CSR-1 protein in somatic blastomeres contributes to the maternal mRNA clearance in these cells.

(3) The authors have not provided data on the quantitative contribution of CSR-1 to mRNA clearance. For example, the authors should show the rate of mRNA degradation in CSR-1 mutants, specifically, in Fig 3a for maternal cleared mRNAs or CSR-1 embryonic mRNA targets, and in Fig 4a for early- or late-degraded mRNAs. These data have already been collected for data shown in Fig 3b. This is a critical point of the paper, and currently unclear whether a small group of maternal mRNAs are degraded mainly by CSR-1 or if CSR-1 is merely one mechanism that participates in clearance of these mRNAs. It seems CSR-1 is only one of the mechanisms for mRNA clearance, as even the highest CSR-1 targeted genes (~115 genes with 22G RNA density >150RPM) on average show less than 2 fold mRNA upregulation in CSR-1 mutant early embryos. Supporting this point, the authors describe a CSR-1 independent mechanism for clearance of late-degraded mRNAs. All this considered, the authors should soften their language in the title/manuscript to for example, "a small RNA pathway that facilitates the clearance of untranslated mRNAs".

We agree with the Reviewer #3 that this is an important point of the paper and we thank the Reviewer #3 for helping us to better present these data in the manuscript. First, we have compared the level of mRNA degradation in CSR-1 depleted embryos or control embryos with the level of maternal inherited mRNAs in 1-cell embryos. We have also calculated the kinetics of the maternal mRNAs and their degradation rate comparing the genes targeted or non-targeted by CSR-1. We found that 1) the cleared mRNAs that have high density of CSR-1 22G-RNAs are inherited in the embryo at much higher level than the non-target mRNAs (Fig. 3b) and their degradation rate is higher (Fig. 3c), and 2) CSR-1 contributes to increase the degradation rates of these targets (Fig. 3d). Indeed, the CSR-1 mRNA targets have similar degradation level than the non-target maternal cleared mRNAs in *csr-1* depleted embryos (Fig. 3d). These results, suggest that in addition to a basal (still unknown) mechanism(s) of mRNA degradation in embryos, that would act on maternal cleared mRNAs targets and non-targets of CSR-1, the CSR-1 pathway contributes to fasten the degradation rate of highly abundant maternal inherited mRNAs. We speculate that the high abundance of CSR-1 mRNA targets in 1-cell embryos requires an additional boost of mRNA degradation to facilitate their clearance in a timely manner.

Finally, we have analyzed how CSR-1 targets early vs late degraded mRNAs. We couldn't evaluate the degradation rate of late degraded mRNAs in our dataset of *csr-1* depleted embryos since the late degraded mRNAs are not yet degraded at the stage we collected CSR-1 depleted embryos. However, we found that early degraded mRNAs are highly enriched for high density of CSR-1 22G-RNAs instead of late degraded mRNAs (Extended Data Fig. 7c), suggesting that CSR-1 preferentially regulates early degraded transcripts. These results are in line with the kinetics

of CSR-1 22G-RNAs across embryonic development (Extended Data Fig. 7d), showing no increase of 22G-RNAs antisense to late degraded transcripts in late embryos, together with the documented degradation of CSR-1 protein at late embryonic stages (previous Extended Data Fig. 1b).

(4) In the title, abstract, and manuscript line 89 (and some other places), the authors state germline inherited small RNAs are important for clearance/ degradation of maternal mRNAs. However, the authors have not provided experimental evidence showing these small RNAs (CSR-1 22G-RNAs) are indeed inherited from the maternal germline. Instead, their data argue the small RNA population is correlated with the translational status of the mRNAs, raising the possibility some of these small RNAs are made de novo in the embryo. The authors should remove “inherited small RNAs” unless new experiments are provided.

We agree with the Reviewer #3 that we have not provided experimental evidences showing the inheritance of CSR-1 22G-RNAs in embryos and that we cannot exclude that CSR-1 is loaded with zygotically-produced 22G-RNAs that are actively synthesized in the early embryos. To test that the 1-cell embryos are already provided with a pool of 22G-RNAs antisense to maternal cleared mRNAs, we have now cloned and sequenced 22G-RNAs in a time-course experiment using sorted population of embryos enriched in 1-cell, 2-cells, 4-50 cells, and > 50 cells. Our results, show that the 1-cell embryos already have abundant 22G-RNAs from CSR-1 targets and their levels do not increase during embryogenesis. Indeed, the 22G-RNA levels remained stable in 1-cell and 2-cells stage embryos and gradually decreased in later stages (Fig. 3e). Therefore, this result demonstrates that the 1-cell embryo already inherit a pool of abundant 22G-RNAs. We also mention in the paper that this experiment does not exclude the possibility that the level of inherited pool of small RNAs can be temporarily maintained by an active mechanism involving the RdRP EGO-1, which is also known to be inherited in embryos (Claycomb et al, 2009).

REVIEWERS' COMMENTS

Reviewer #3 (Remarks to the Author):

In the revised manuscript, Quarato et al. provide clarifications to some of my comments. I think this manuscript is of high quality and the findings provide new insight toward our understanding of the functions of small RNAs in mRNA degradation and its relationship to the translation status of the mRNAs. Still, I believe the presentation of the key data on the roles of CSR-1 in degrading maternal RNAs can be improved. While the model in Fig 7 clearly presents the difference in CSR-1 target mRNA levels between wild type and CSR-1-slicer mutant, the data in Figure 3d and Figure 5a are not presented in a similar format to clearly address the contribution of CSR-1 in mRNA clearance. This is a critical point as I suspect such analyses may suggest a distinct model (see below). Since only the mRNAs highly targeted by CSR-1 exhibit significant change in rate of degradation, the author should at least compare the degradation kinetics of those genes between wild type and CSR-1 depletion in the format presented in Figure 3B.

Since the authors data showed that CSR-1 slicer activity modulates gene expression with similar amplitude and efficiency in the germline as in the embryo, those CSR-1 target mRNA transcripts are likely already downregulated by CSR-1 in germline/oocytes and continue to be degraded by CSR-1 in early embryos. Indeed, a previous study (Adina Gerson-Gurwitz et al., 2016) has shown that CSR-1 slicer activity represses the expression of germline transcripts and affects the cell division of one-cell embryo. Therefore, the author may see an increase of CSR-1 target mRNAs inherited in one-cell embryos in *csr-1* depletion and those mRNAs continue to be degraded at a slower rate in *csr-1* depleted embryos. If so, the author should adjust the model accordingly.

Another minor suggestion for the authors is to change "fasten" to "quicken", as quicken but not fasten means to speed up the process.

Taken together, I think the manuscript (with the suggested modifications), is suitable for publication at Nature Communications.

REVIEWER COMMENTS

Reviewer #3 (Remarks to the Author):

In the revised manuscript, Quarato et al. provide clarifications to some of my comments. I think this manuscript is of high quality and the findings provide new insight toward our understanding of the functions of small RNAs in mRNA degradation and its relationship to the translation status of the mRNAs.

We thank the Reviewer #3 for appreciating the high quality of our revised work and our novel findings.

Still, I believe the presentation of the key data on the roles of CSR-1 in degrading maternal RNAs can be improved. While the model in Fig 7 clearly presents the difference in CSR-1 target mRNA levels between wild type and CSR-1-slicer mutant, the data in Figure 3d and Figure 5a are not presented in a similar format to clearly address the contribution of CSR-1 in mRNA clearance. This is a critical point as I suspect such analyses may suggest a distinct model (see below). Since only the mRNAs highly targeted by CSR-1 exhibit significant change in rate of degradation, the author should at least compare the degradation kinetics of those genes between wild type and CSR-1 depletion in the format presented in Figure 3B.

We agree with the Reviewer #3 that it would be interesting to measure the degradation rate of maternal mRNAs in wild-type and *csr-1* depleted embryos at 1-cell stage embryos, early embryos and late embryos as presented in Figure 3b for wild-type embryos. However, the results presented in Figure 3b have been obtained using a FACS sorting strategy (explained in the methods) coupled with RNA-seq exclusively in a wild-type background and not in *csr-1* depleted embryos. It was not feasible in fact to combine the *csr-1* depletion strain with the transgenic strain used for FACS sorting. This is why we are only comparing wild-type and *csr-1* depletion (or catalytic dead) in a single timepoint (mixed early staged embryos) in Figure 3d and Figure 5d and we cannot change their representation as in Figure 3b.

Since the authors data showed that CSR-1 slicer activity modulates gene expression with similar amplitude and efficiency in the germline as in the embryo, those CSR-1 target mRNA transcripts are likely already downregulated by CSR-1 in germline/oocytes and continue to be degraded by CSR-1 in early embryos.

Indeed, a previous study (Adina Gerson-Gurwitz et al., 2016) has shown that CSR-1 slicer activity represses the expression of germline transcripts and affects the cell division of one-cell embryo. Therefore, the author may see an increase of CSR-1 target mRNAs inherited in one-cell embryos in *csr-1* depletion and those mRNAs continue to be degraded at a slower rate in *csr-1* depleted embryos. If so, the author should adjust the model accordingly.

The data presented in this manuscript together with the data presented in Singh et al. BioRxiv 2020 showed that CSR-1 slicer activity modulates target transcripts with similar amplitude and efficiency in the germline as in the embryo. However, the majority of mRNAs highly targeted by CSR-1 in embryos are not highly targeted by CSR-1 in adult germlines (Figure 2a). Because only the mRNAs highly targeted by CSR-1 exhibit significant change in rate of degradation, CSR-1 embryonic targets are preferentially downregulated by CSR-1 in embryos and not in adult germlines. For this reason, we propose that the majority of maternal-inherited embryonic CSR-1 targets are degraded in the embryos. Nonetheless, we cannot exclude that some targets might start to be regulated by CSR-1 in oocyte as soon as translation decreases (we have discussed this in the manuscript).

Another minor suggestion for the authors is to change “fasten” to “quicken”, as quicken but not fasten means to speed up the process.

We have changed the word fasten to quicken according to the Reviewer #3 suggestion.

Taken together, I think the manuscript (with the suggested modifications), is suitable for publication at Nature Communications.